# Essential gene prediction using limited gene essentiality information–An integrative semi-supervised machine learning strategy

**Sutanu Nandi[1,2], Piyali Ganguli[1,2], Ram Rup Sarkar**[1,2]*

**1** Chemical Engineering and Process Development, CSIR-National Chemical Laboratory, Pune, Maharashtra, India, **2** Academy of Scientific & Innovative Research (AcSIR), Ghaziabad, India

* rr.sarkar@ncl.res.in

**Data Availability Statement:** All relevant data are within the manuscript and its Supporting Information files.

## Abstract

Essential gene prediction helps to find minimal genes indispensable for the survival of any organism. Machine learning (ML) algorithms have been useful for the prediction of gene essentiality. However, currently available ML pipelines perform poorly for organisms with limited experimental data. The objective is the development of a new ML pipeline to help in the annotation of essential genes of less explored disease-causing organisms for which minimal experimental data is available. The proposed strategy combines unsupervised feature selection technique, dimension reduction using the Kamada-Kawai algorithm, and semi-supervised ML algorithm employing Laplacian Support Vector Machine (LapSVM) for prediction of essential and non-essential genes from genome-scale metabolic networks using very limited labeled dataset. A novel scoring technique, Semi-Supervised Model Selection Score, equivalent to area under the ROC curve (auROC), has been proposed for the selection of the best model when supervised performance metrics calculation is difficult due to lack of data. The unsupervised feature selection followed by dimension reduction helped to observe a distinct circular pattern in the clustering of essential and non-essential genes. LapSVM then created a curve that dissected this circle for the classification and prediction of essential genes with high accuracy (auROC > 0.85) even with 1% labeled data for model training. After successful validation of this ML pipeline on both Eukaryotes and Prokaryotes that show high accuracy even when the labeled dataset is very limited, this strategy is used for the prediction of essential genes of organisms with inadequate experimentally known data, such as *Leishmania sp*. Using a graph-based semi-supervised machine learning scheme, a novel integrative approach has been proposed for essential gene prediction that shows universality in application to both Prokaryotes and Eukaryotes with limited labeled data. The essential genes predicted using the pipeline provide an important lead for the prediction of gene essentiality and identification of novel therapeutic targets for antibiotic and vaccine development against disease-causing parasites.

**Funding:** We thank SERB, Department of Science and Technology, Govt. of India (DST/ICPS/EDA/2018) and DBT, Department of Biotechnology, Govt. of India (File No. BT/PR14958/BID/7/537/2015), for providing financial support to Ram Rup Sarkar. Sutanu Nandi acknowledges DST-INSPIRE for Senior Research Fellowship. Piyali Ganguli acknowledges the Council of Scientific & Industrial Research (CSIR) for the Senior Research Fellowship. The funders had no role in study design, data collection and analysis, decision to publish, or preparation of the manuscript.

**Competing interests:** The authors have declared that no competing interests exist.

## 1. Introduction

Gene essentiality information of disease-causing organisms that throws light on the minimally essential genes that are absolutely required for the survival of the organism under any environmental condition has not only been indispensable for the prediction of novel therapeutic targets for antibiotic and vaccine development but has also contributed towards industrial bioprocessing, food microbiology, and bioremediation. However, experimental techniques [1–5] like genetic foot-printing, gene knockouts, RNA interference (RNAi), transposon mutagenesis have been employed to perform a genome-wide screen to check for gene essentiality are expensive, labor-intensive, as well as time-consuming.

As an efficient alternative to these highly complex experimental strategies, researchers now are employing computational techniques based on homology mapping, constraint-based modeling strategies, and machine learning strategies [6–8]. The homology-based essential gene prediction methods rely on the fact that essential genes are less likely to evolve, tend to remain conserved, and are often shared by distantly related organisms. Essential genes have been identified by comparative genomic analysis in different bacterial species such as Mycoplasma [9], *Liberibacter* [10], *Plasmodium falciparum* [11], and *Brucella spp.* [12]. However, the limitation of this method is that the conserved ortholog genes between different species form only a small fraction of the entire genome [13]. Also, it has been observed that highly conserved genes across different species are not always essential, as gene essentiality also depends on different environmental conditions where the organism resides.

Constraint-based modeling strategies, such as Flux Balance Analysis (FBA), employ genome-scale reconstructed metabolic networks to predict the metabolic fluxes at steady-state. This methodology is widely used for predicting essential genes by performing *in-silico* knock-out of a gene and estimating its corresponding lethality [14–16]. A limitation of this FBA method is that only a limited number of environmental conditions can be considered for a certain biomass equation (or objective function) with respect to gene essentiality.

On the other hand, Machine Learning (ML) strategies comprise various data-driven approaches that train a model from the inherent patterns of the training data and make a prediction for the unlabeled data. These ML algorithms can be broadly grouped under supervised, semi-supervised, and unsupervised strategies [17,18]. The supervised strategies such as Decision Tree, Naïve Bayes, Support Vector Machine (SVM), etc. require sufficient amounts of labeled data for model training. In contrast, the unsupervised method relies on clustering algorithms (e.g., K-Means Clustering), where no labeled data is required. The semi-supervised ML algorithms that comprise Generative Models, Self-Training, Transductive SVM, and Laplacian SVM combine the potential of both supervised and unsupervised ML strategies and can train the model with a very limited amount of labeled data. At the same time, optimization of the hyper-parameter is crucial for enhancing the predictive performance of these machine learning classifiers. Various meta-heuristic techniques, such as Particle Swarm Optimization (PSO) [19], Genetic Algorithm (GA) [20], Ant Colony Optimization (ACO) [21], Grey Wolf Optimizer (GWO) [22], Ant Lion Optimizer (ALO) [23], etc. have been used for hyper-parameter tuning.

Based on the availability of labeled data of essential genes, researchers have employed supervised machine learning strategies [6–8] as well as deep learning-based strategies to predict essential genes [24,25]. The key advantage of these strategies lies in the fact that these models are capable of capturing the inherent patterns of a large array of biologically relevant 'features' that are distinctive and reflect the heterogeneous properties of essential genes. Supervised machine learning classifiers such as logistic regression [26,27], support vector machine [28–31], random forest [32], decision tree [26], ensemble [26] and probabilistic Bayesian-based

methods [26,27,33] and instance-based learning methods such as K Nearest neighbor (K-NN) and Weighted KNN (WKNN) [34] have been used for gene essentiality prediction. Deep Learning strategies based on multilayer perceptron networks have also been used for essential gene prediction [24,35]. In these studies, researchers have mostly opted for simpler optimization methods for parameter tuning, such as the grid search technique, where the entire parameter space is explored in all possible combinations.

These machine learning-based classifiers predict gene essentiality of unannotated genes based on the pattern of the features of previously annotated genes that have been verified experimentally and labeled as essential and non-essential. In order to achieve this, researchers have curated different combinations of features. Most of the machine learning approaches use calculated features either from coding sequences [36–38] or network (e.g., protein interaction network, metabolic network) topological features [6,39] or both. Features, such as amino acid frequency and protein length computed from protein sequence, and codon adaptation index (CAI), Effective Number of Codons (ENC), Phyletic Retention (PR), GC content computed from nucleotide sequence are some of the known features of gene essentiality across bacteria [28,29,40]. Protein interaction networks (PIN) have been used to calculate topological network features to classify gene essentiality [28,39]. However, these strategies fail for many organisms that do not hold the idea of the centrality-lethality hypothesis in a PIN [41]. On the other hand, few studies have used flux-based features derived from metabolic networks to classify genes [29,30] that have been calculated under a single environmental condition that does not represent a universal set of features. Detailed reviews of the existing machine learning strategies for gene essentiality prediction have been discussed in different works of literature [6–8].

A major drawback of these existing machine learning algorithms for essential gene prediction is that they require a large amount of these labeled data that helps to train these models for an accurate prediction of the essentiality of unannotated genes, and show very poor performance when the labeled data set is imbalanced or limited. To circumvent these problems, in our previous study, an integrative machine learning strategy has been developed using a combination of feature selection algorithm, Support Vector Machine- Recursive Feature Elimination (SVM-RFE) [42] and classifier, Sequential Minimal Optimization (SMO) [43] for gene essentiality prediction in the metabolism of *Escherichia coli*, which performed well on imbalanced data set with diverse features computed from flux coupled connected sub-network along with other sequence-based features [40]. Here, the advantages of using the Flux Coupling Analysis (FCA) based feature for the prediction of gene essentiality with high accuracy and confidence have been reported. FCA analysis help to capture the physiological dependence of one gene-reaction combination on another, which is coupled to it, under all input exchanges of a reaction, representing all possible environmental conditions, thereby helping the classifier to accurately identify the minimally essential genes that are absolutely crucial for sustaining the metabolic demands of the cell to ensure its survival [40]. However, this technique was unable to predict gene essentiality when a very small amount of experimentally verified labeled data are available.

To mitigate the problems inherent in the existing strategies, we propose an integrative semi-supervised machine learning strategy based on Laplacian SVM [44] for the classification of genes using gene sequence, protein sequence, network topological, and flux-based features with very limited labeled data on gene essentiality of metabolic networks for both Prokaryotic and Eukaryotic organisms. Another objective of this work is the development of a new machine learning pipeline to help in the annotation of essential genes of less explored organisms, like *Leishmania donovani* and *Leishmania major*, the causative organisms for the neglected tropical disease Leishmaniasis, for which very limited experimental data is available. By using the available tools and techniques, the prediction of gene essentiality and targeted

therapy for the disease becomes extremely difficult [45]. In the present work, it is hypothesized that using these diverse features, like topological network features of both the genome-scale metabolic reaction network as well as the flux-coupled sub-networks, together with the sequence-based features simultaneously, that can capture both the properties of genotype and phenotype and by employing the proposed algorithm, it is possible to predict the essentiality of uncharacterized genes with high accuracy even in the cases where labeled data is limited. This is in contrast to other machine learning pipelines for essential gene prediction that relies on only sequence-based features and has been applied to only Prokaryotes [26,46]. In this work, the novel features derived from the genome-scale metabolic reaction network, as well as the flux-coupled sub-networks, contribute towards the better prediction of gene essentiality by capturing the contribution of a gene in sustaining the metabolic demands of the cell under varied environmental challenges that are indispensable for its survival. A new scoring technique has also been proposed, called the Semi-Supervised Model Selection Score (SSMSS) that correlates well with Mathews Correlation Coefficient (MCC) [47] and can be used for the selection of the best model when the calculation of supervised performance metrics like MCC or auROC is difficult due to lack of experimental data. After the successful validation of this proposed pipeline on twelve organisms, with well-annotated genes essentiality information, using as low as 1% labeled data on two types of training datasets (i.e., with 80% training and 20% blind datasets, as well as using the whole dataset for training), the essential genes in *Leishmania* have been predicted as well as categorized the reaction-gene pairs in five different groups based Gene-Protein-Reaction (GPR) association in metabolism. These groups depict the association of the reactions with different combinations of essential and non-essential genes, which throws light on the probable reaction-gene combination that can be used for targeted therapy. This study promises to lay the foundation to the prediction of gene essentiality information for less explored organisms that will help experimental biologists to identify novel therapeutic targets even when only limited information is available.

## 2. Methods

The Machine learning strategy developed to predict gene essentiality, as elucidated in Fig 1, combines feature selection technique based on a space-filling concept, dimension reduction (DR) using the Kamada-Kawai (KK) algorithm, and classification of genes based on a semi-supervised machine learning algorithm employing Laplacian Support Vector Machine (LapSVM). This pipeline combines heterogeneous biological features, such as sequence-based, as well as network-based features. It classifies genes based on a training dataset of very limited information of essential genes from experimental data. Twelve organisms comprising of both Prokaryotes and Eukaryotes (Table 1) with well-annotated genes essentiality information from the OGEE database [48] have been considered for the validation of this proposed strategy, and the subsequent prediction of essential genes in *Leishmania major* and *Leishmania donovani* have been performed. The gene essentiality information has only been considered from the OGEE database as this collates data using text mining as well as manually verified with experimental data, unlike other gene essentiality databases that rely on only text mining.

### 2.1 Training data and Testing data set preparation and integration of heterogeneous features

The training datasets for the pipeline of the 12 target organisms were prepared by calculating mainly two types of features: topological features and sequenced based features. These features were extracted primarily from the genome-scale reconstructed metabolic networks, the fasta files containing the coding nucleotide sequences of the genes, and protein sequences of these

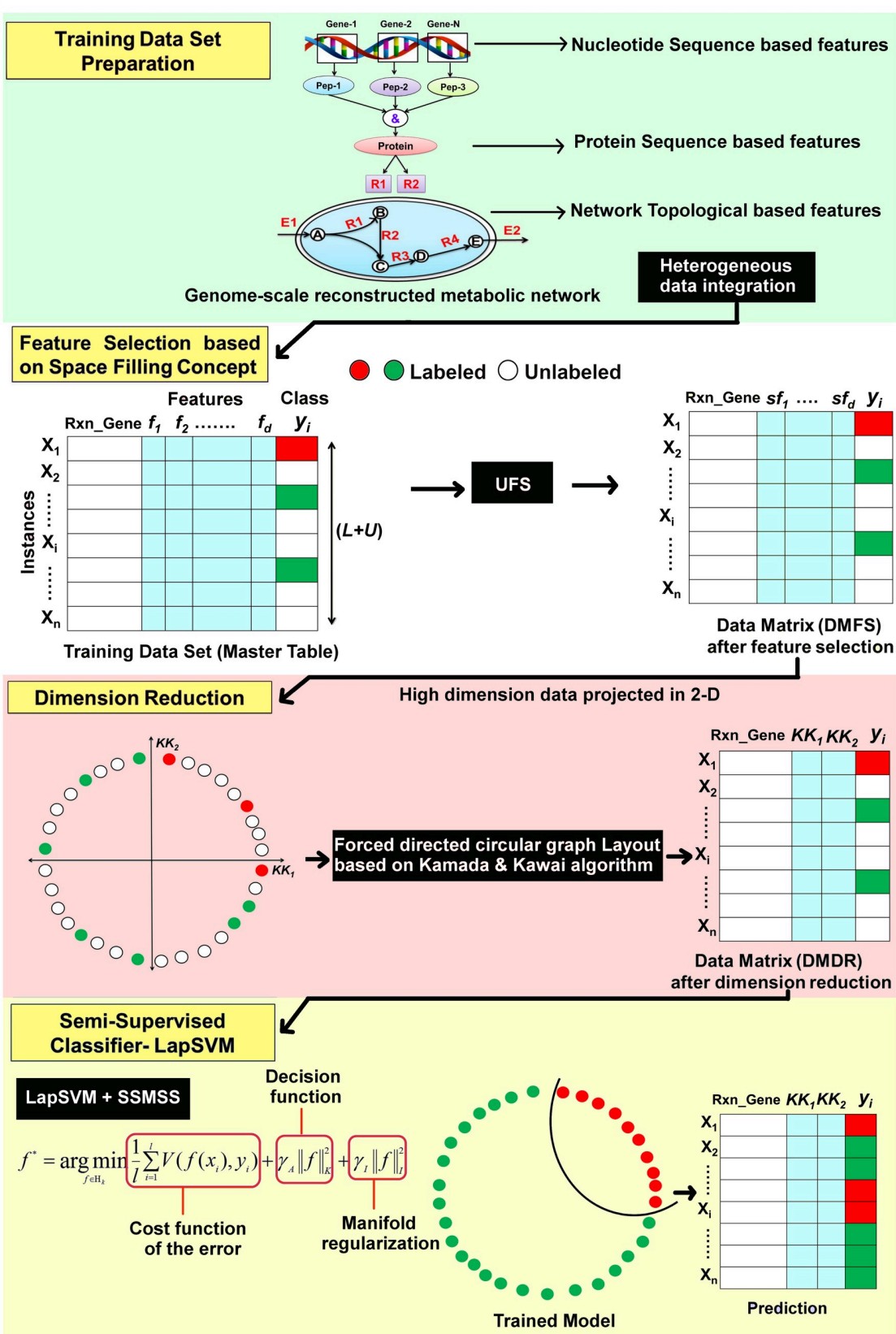

**Fig 1. The proposed machine learning strategy.** The integrated pipeline for prediction of essential genes based on limited labeled training dataset consisting of reaction-gene pairs with sequence, informatics, and topological network features.

target organisms (Table 1) [62]. From the genome-scale reconstructed metabolic network, the information of metabolites, reactions, and genes was collated.

The sequence-based features and the topological features of the metabolic reaction network, and flux-coupled sub-network based were calculated and accumulated for each reaction-gene combination. These reaction-gene combinations integrate diverse features of the metabolic adaptation of the organism and give detailed insights into the role of a particular gene in the metabolic reaction network. This helps in the prediction of the essentiality of the gene in the target organism with high accuracy. A total of 289 features were computed for each reaction-gene pair. Brief descriptions of these features are given below, and their abbreviation**s** are enlisted in S1 Table.

To establish the model consistency and reproducibility of the proposed pipeline, two different types of data sets for each of the twelve organisms have been used. The first type of data set consists of 80% data points of total data set with limited labeled data that is used for training while the remaining 20% is used for blind testing to check the model validation. Using this 80% data points of the whole dataset, different types of training data set are further created with limited labeled data points in the range, i.e., i % Labeled (L) and (100—i%) Unlabeled (UL) data, where i = 1, 2, 3, 4, 5, 10, 30, 50, 70 and 90. In each category, labeled samples were chosen randomly from the master table. It is to be mentioned here that this selection of labeled data was conditionally randomized to ensure that both the essential and non-essential genes categories appear with equal probability. In this way, 100 data sets in each labeled category have been created.

The second type of data set consists of the whole dataset with limited labeled data, which is used for model training and prediction purposes for each of the twelve organisms. It is to be

**Table 1. Organisms considered for model training and validation.**

| Organism Name | Abbreviation | Input files | |
|---|---|---|---|
| | | FASTA files of coding nucleotide and protein sequence (RefSeq assembly accession) | Genome-Scale Reconstructed Metabolic Network |
| **Organisms used for Model Development and Validation of the Proposed Pipeline** | | | |
| *Acinetobacter sp.* ADP1 | ACIAD | GCF_000046845.1_ASM4684v1 | iAbaylyiv4 [49] |
| *Bacillus subtilis subsp. subtilis str.* 168 | BACSU | GCF_000009045.1_ASM904v1 | iYO844 [50] |
| *Escherichia coli* K-12 MG1655 | ECOLI | GCF_000005845.2_ASM584v2 | iJO1366 [51] |
| *Helicobacter pylori* | HELPY | GCF_000008525.1_ASM852v1 | iIT341 [52] |
| *Mycobacterium tuberculosis* H37Rv | MYCTU | GCF_000195955.2_ASM19595v2 | iNJ661 [53] |
| *Pseudomonas aeruginosa* PAO1 | PSEAE | GCF_000006765.1_ASM676v1 | iPae1146 [54] |
| *Pseudomonas aeruginosa* UCBPP-PA14 | PSEAB | GCF_000014625.1_ASM1462v1 | iPau1129 [54] |
| *Salmonella enterica subsp. enterica serovar Typhimurium* LT2 | SALTY | GCF_000006945.2_ASM694v2 | STM_v1_0 [55] |
| *Staphylococcus aureus subsp. aureus* NCTC 8325 | STAAB | GCF_000013425.1_ASM1342v1 | BMID000000141098 [56] |
| *Saccharomyces cerevisiae* | YEAST | GCF_000146045.2_R64 | iMM904 [57] |
| *Caenorhabditis elegans* | CELEG | GCF_000002985.6_WBcel235 | iCEL1273 [58] |
| *Mus musculus* | MUSMU | GCF_000001635.26_GRCm38.p6 | iMM1415 [59] |
| **Organisms used for Case Study** | | | |
| *Leishmania donovani* | LDONO | TriTrypDB-36 | iMS604 [60] |
| *Leishmania major* | LMAFR | TriTrypDB-36 | iAC560 [61] |

mentioned here that, in less-studied organisms where gene essentiality information is very less, a blind test cannot be applied. For those cases, the whole data set with limited labeled data will be used for model training and prediction purposes.

**Topological analysis of reaction and flux-coupled sub-network.** The metabolic network of each target organism was transformed into an undirected reaction network (RN), in which each node denotes an enzyme (reaction), and each edge represents the connection between two reactions that have common metabolites. The commonly used topological network features, such as centrality measures, that highlight the biological significance of an enzyme in a network were computed [63]. Generally, a central and highly connected enzyme in biological networks is often essential as it represents an important hub within the network [64]. If this hub node is blocked, then the whole pathway might be disrupted.

Similarly, Flux coupling analysis (FCA) is an optimization procedure based on flux, which represents whether the reaction subsets are coupled or not in certain given specific environmental exchange constraints [65,66]. Flux-coupled subgraph was used to extract biologically relevant topological features dependent on physiological flux relationships.

Eight centrality measures have been computed for both the reaction as well as the flux coupled networks, *viz.*, Degree Centrality, Eigenvalue Centrality, Eccentricity, Hub score, Authority Scores, Page Rank, Betweenness Centrality, and Number of triangles. A detailed description of all these centrality measures has been discussed in different literature [67–69]. These topological features have been calculated using the "igraph" package in R [70].

**Features derived from the coding nucleotide sequence.** Three types of features (*viz.* nucleotide content, codon usage bias, and information-theoretic features) of the metabolic genes have been extracted from the nucleotide sequence of the organisms that contribute towards gene essentiality. A brief description of the features has been discussed below.

**Nucleotide content.** Previous studies have elucidated that in bacterial genomes, GC content is correlated with the environmental condition in which the bacterium survives [71]. Hence, the related GC content of the genome of a target organism can be an essential feature for gene essentiality prediction. Another study showed that there is a significant difference in the distribution of the frequency of occurrence of A, T, G, and C nucleotides at the 3rd synonymous position of codons between the essential and non-essential genes [40]. These features were computed using an in-house code.

**Codon usage bias.** Protein abundance in an organism can be predicted by using Codon usage [72–74]. Highly expressing abundant proteins in metabolism might have functional importance and can be essential. Codon usage bias features, like Effective Number of Codons (ENC) [75] and Codon Adaptation Index (CAI) [73], were calculated using EMBOSS package version 6.6.0–1 [76].

**Mutual Information (MI) and Conditional Mutual Information (CMI).** A previous study has used information-theoretic features such as mutual information (MI) and conditional mutual information (CMI), for essential gene prediction [37]. MI and CMI profile of coding nucleotide sequence can be used as genomic signatures which represent the phylogenetic relationship between genomic sequences [77]. A total of 80 features (16 MI and 64 CMI) have been computed by using in house Perl script.

**Features derived from protein sequence.** In order to investigate the dependence of gene essentiality on protein sequences, various derived and informatic features such as the frequencies of the amino acids, protein length, paralogy score, average Kidera factor, etc. have been considered in this study.

**Frequencies of the twenty amino acids and protein length.** Each protein sequence related to the reaction-gene combination was used to calculate the occurrences of the 20 amino acids that reflect the physicochemical properties of these proteins related to each of the

reaction-gene combinations under consideration. These twenty features were calculated using EMBOSS package version 6.6.0–1 [76] and named according to their corresponding 20 amino acids.

**Paralogy based features (paralogy score).** The sequence similarity of a gene in its intragenome is called a paralogous gene of an organism. Paralogous genes have the same or similar types of biological functions. An organism may not be affected by the deletion of one of the paralogous genes because another paralogous gene may compensate for a similar type of function. So there are fewer chances for paralogous genes to be essential [78].

The paralogy score of a gene was calculated by performing a BLAST [version 2.2.26] search against the whole set of protein sequences of a target organism with different E-value threshold ranging from $10^{-3}$ to $10^{-30}$ with at least 40% identity. Features based on paralogy score were labeled as P3 (E-value cut off $10^{-3}$), P5 (E-value cut off $10^{-5}$), P7 (E-value cut off $10^{-7}$), P10 (E-value cut off $10^{-10}$), P20 (E-value cut off $10^{-20}$), P30 (E-value cut off $10^{-30}$). These features have been calculated using in house Perl script.

**Fourier sine and cosine coefficient.** The Fourier sine and cosine coefficient of protein sequences [79] have been used to see if there are any inherent patterns which will help to classify between essential and non-essential genes. The Fourier coefficient (FC) is the converted numerical values of protein sequences, which describes the physical properties of corresponding amino acids. These physical properties represent the ten property factors using factor analysis introduced by Kidera et al. [80]. Mathematical representations of these coefficients are given below:

$$FC\sin WN_k\_KF_n = a_k^{[n]} = \sum_{l=0}^{N-1} f_l^{[n]} \sin\left(\frac{2\pi kl}{N}\right) \qquad (Eq\ 1)$$

$$FC\cos WN_k\_KF_n = b_k^{[n]} = \sum_{l=0}^{N-1} f_l^{[n]} \cos\left(\frac{2\pi kl}{N}\right) \qquad (Eq\ 2)$$

Where the length of the protein sequence is N, $f_l^{[n]}$ is n[th] property factor of amino acid *l*, and wavenumber is *k* (Eqs 1 and 2).

Fourier sine and cosine coefficient in a specific range of Wave Number (WN) and Kidera Factor (KF) was calculated. The range of WN and KF are $0 \leq k \leq 7$ and $1 \leq n \leq 10$. It is also reported that global folding information of the protein is encoded in a specific range of wavenumber $0 \leq k \leq 7$ [79]. A total of 150 features were computed. These features have been calculated using in house Perl script.

**Average Kidera factor.** The ten Kidera Factors (viz. KF1: Helix/bend preference, KF2: Side-chain size, KF3: Extended structure preference, KF4: Hydrophobicity, KF5: Double-bend preference, KF6: Partial specific volume, KF7: Flat extended preference, KF8: Occurrence in the alpha region, KF9: pK-C, KF10: Surrounding hydrophobicity) were derived by multivariate analysis on 20 amino acids using 188 physical properties and dimension reduction techniques [80]. The protein sequence of the corresponding reaction-gene combination was used to calculate ten features ($AKF_i$ where, i = 1 to 10) by averaging the ten Kidera factors. These features have been calculated using in house Perl script.

## 2.2 Feature selection based on the space-filling concept

The contribution of these 289 features towards gene essentiality is unknown; hence, there may be a possibility to select redundant features by the feature selection algorithm. These redundant features may affect the training performance of the machine learning model. Hence, it is

important as well as challenging to choose the non-redundant, unique feature subset for training the model. Feature selection helps to capture the most relevant biological features and helps the classifier to learn a better way to predict essential and non-essential genes with high accuracy. Here the unsupervised feature selection method based on the space-filling concept has been used [81]. This unsupervised method selects the features based on a coverage measure that estimates the spatial distribution of the data points in a hypercube and ensures uniform distribution of points in a regular grid in the data space. The method captures the variability of features with new and relevant information about the data. This method has been tested on various datasets and different scenarios with noise injection and data shuffling. The benefits of using this algorithm are two folds. Firstly, being an unsupervised algorithm, prior information of the output variable is not required.

Additionally, here no classifier is required for feature selection. Hence time complexity is less in comparison to other feature selection algorithms, like SVM-RFE. Also, it has been observed that this method gives better information of relevant features than other unsupervised correlation-based feature selection techniques that, although it can remove the redundant features, cannot eliminate the features with low variability that are non-relevant and non-informative for classification [82,83].

## 2.3 Dimension reduction using forced directed graph layout

After feature selection, the data set was transformed into a lower dimension (2-D) using a dimension reduction technique for visualization. Projected 2-D features set to reserve all the information the same as higher-dimensional data. This is an important step in the pipeline as the classifier works better in 2-D than with the higher dimension data. For dimension reduction, a force-directed graph layout algorithm Kamada-Kawai has been used that considers each data point as a node in a graph having attractive and repulsive forces between them that can be modeled as springs connecting the nodes [84]. The algorithm then tries to cluster the data points by minimizing the total energy of the system based on attracting and repelling forces between them. Here the input of the Kamada-Kawai algorithm is a graph constructed by using the K-Nearest Neighbour (K-NN) algorithm. For known organisms, it has been observed that essential genes are clustered together in one side of an arc in a circle layout, and non-essential genes are clustered in the rest of the circle. A circular layout of each organism has been observed from the Kamada Kawai algorithm with a specific parameter (K Nearest Neighbor) value of the K-NN algorithm. Here it is assumed that if a similar circular layout is observed for less explored organisms related to gene essentiality, the unlabeled genes will be clustered together category wise and reside on the arc of the circle. This analysis had been performed using the "dimRed" package in R [85].

Both the feature selection and the dimensionality reduction methods are used for not only reducing the number of features in a dataset but also to select the important features, which are contributing significantly. Feature selection is used for selecting the relevant features without changing the original values, whereas, the dimensionality reduction step transforms the higher dimensional features into a lower dimension. From the dimension reduction technique it is very difficult to identify the key features which are contributing for classifications, hence the feature selection step is necessary.

To test the efficiency of this dimension reduction technique combined with unsupervised feature selection and LapSVM classifier, the performance metrics of Kamada-Kawai has been compared against other dimension reduction techniques, such as Principal Component Analysis (PCA) [86], Metric Dimensional Scaling (MDS) [87], Fruchterman Reingold [88] and FastICA [89] using the gold standard dataset of twelve organisms. To test the statistical

significance of the results, the one-tailed Mann Whitney U Test has been performed with 1% level of significance ($P<0.01$).

## 2.4 Semi-supervised classifier: Laplacian SVM

Essential gene classification using the machine learning technique can be a difficult task when a minimal amount of gene essentiality information for the target organism is available. In this setting, semi-supervised learning is an appropriate approach that builds a trained model from labeled and unlabeled samples [90]. Most of these semi-supervised algorithms follow two common assumptions, i.e., cluster assumption and manifold assumption. Cluster assumption states that data points in the same cluster have a chance of having the same class label. Manifold assumption means that close data points along the manifold area follow similar data structures or similar class labels. However, cluster assumption follows the global feature, and manifold assumption follows the local features in the model.

Laplacian support vector machine (LapSVM) is a graph-based semi-supervised learning method, which is based on a manifold regularization framework [44]. The graph is constructed from labeled and unlabeled data as the node. The similarity between data points in a graph can be assigned by edge weight, which is calculated from the K-NN algorithm. In this way, the information of labeled data points can be passed to another node, and then, the unlabeled nodes can be labeled. The input data set being circular (non-linear), Radial Basis Function (RBF) kernel with the classifier LapSVM have been used. This analysis had been performed using the "RSSL" package in R [91].

## 2.5 The score for best model selection

There are various performance metrics, e.g., True Positive Rate (TPR), False Positive Rate (FPR), precision, recall, F-measure, Matthews correlation coefficient (MCC), Area under the receiver operating characteristic curve (auROC), etc. to evaluate the trained model in supervised machine learning technique. These measures are statistically significant if sufficient labeled data are available. However, due to limited labeled data, these metrics will not work for best model selection in a semi-supervised type algorithm. To circumvent the above problem, a new measure has been proposed, called the Semi-Supervised Model Selection Score (SSMSS), for selecting the best model. This SSMSS score is dependent on four different measurements (Eq 3). For this, the training data set, having limited labeled reference, has been labeled as ground truth (GT) reference. Another reference set called the pseudo reference (PR) has been considered by calculating the distance from unlabeled data points to the labeled dataset. The dataset containing the predicted labels by the Laplacian SVM classifier has been labeled as the Laplacian Reference (LR). Thereafter, Silhouette Index (SI) [92] was computed to check the clustering grouping quality. The CorrectPrediction$_{GT\_LR}$ measure was calculated based on the matches between the predictions of the Laplacian SVM classifier with the Ground Truth data. Here, the calculation of the MCC with the help of Pseudo-reference and Laplacian Reference was represented as MCC$_{PR\_LR}$. Silhouette Index calculation based on Pseudo Reference and Laplacian Reference was denoted by SI$_{PR}$ and SI$_{LR}$ respectively. Based on these parameters, the values of the proposed Semi-Supervised Model Selection Score (SSMSS) may vary from 0 to 1. If any of the above four measurements is low, then the SSMSS value will be drastically decreased. The best model will be selected from 64 models which has the highest SSMSS value for each data set in different parameters combinations, i.e., kernel parameter [Radial Basis Function (RBF) kernel parameter sigma ($\sigma$)] and LapSVM parameters [lambda ($\lambda$): L$_2$ regularization parameter and gamma ($\gamma$): the weight of the unlabeled data]. It may be mentioned here that the score will not consider those models which have negative Silhouette Index and MCC

value. The parameters $(\sigma, \lambda, \gamma)$ have been varied with four different values, i.e., 0.01,0.1,1,10. Therefore, by tuning these model parameters using grid search, 64 models for each data set have been generated. The following equation has been proposed for the calculation of the SSMSS.

$$\text{SSMSS}_{k=1 \text{ to } 64} = \min\{\text{CorrectPrediction}^k_{GT\_LR}, \text{MCC}^k_{PR\_LR}, \text{SI}^k_{PR}, \text{SI}^k_{LR}\} \qquad \text{(Eq 3)}$$

$$\forall \text{MCC}^k_{PR\_LR} \geq 0, \text{SI}^k_{PR} \geq 0, \text{SI}^k_{LR} \geq 0.$$

$$\text{SSMSS}_{best} = \max\{\text{SSMSS}_{k=1}, \text{SSMSS}_{k=2}, \ldots\ldots\ldots, \text{SSMSS}_{k=64}\},$$

where k is the $k^{th}$ model with a particular parametric combination and $\text{SSMSS}_{best}$ is the best score of the best model among these 64 models.

## 2.6 Time complexity of the proposed strategy

The proposed pipeline has three components (i.e., Unsupervised Feature Selection, Kamada Kawai Dimension Reduction Technique, and LapSVM semi-supervised classifiers), which work sequentially. To calculate the total time complexity T(n,d) of the proposed strategy, the cumulative effect of all three components have been considered, where n denotes the number of data points (reaction-gene pair) that depends on the size of the metabolic network of the organism, and d is the total number of features.

The time required for each of the three components can be represented as follows [44,81,84]:

Time required for Unsupervised Feature Selection algorithm $= \frac{d(d+1)n^2}{2}$ Time required for Kamada Kawai algorithm $= n^3$

Time Required for LapSVM $= n^3$

Therefore, the total time required T(n,d) can be represented as:

$$\therefore T(n, d) = \frac{d(d+1)n^2}{2} + n^3 + n^3$$
$$or, \quad T(n, d) \leq 4n^3 + n^2(d^2 + d)$$
$$or, \quad T(n, d) \leq (4 + d + d^2)n^3$$
$$or, \quad T(n, d) \leq Cd^2n^3$$
$$or, \quad T(n, d) = \mathrm{O}(d^2n^3)$$

Where, $C$ is a constant, in particular, $C \geq 6 \ \forall d, n \in \mathbb{N}$.

Therefore, the total time complexity of the proposed strategy is $O(d^2n^3)$.

## 2.7 Gene essentiality prediction, experimental validation, and pathway enrichment

The essential gene prediction results for the twelve model organisms have been compared with experimental data obtained from the OGEE database, and the corresponding supervised performance metrics such as TPR, FPR, MCC, auROC, etc. were calculated. Further, the predicted essentiality information of the reaction-gene pairs of all twelve organisms has been categorized into five different groups based on their involvement in different reactions. These five groups are following: **CEN** (Combination of Essential and Non-essential), involving both essential and non-essential genes controlling a reaction; **ME** (Multiple Essential), multiple essential genes involved in a reaction; **MN** (Multiple Non-essential), multiple non-essential genes

governed a reaction; **SE** (Single Essential), single essential genes involved in a reaction; **SN** (Single Non-essential), single non-essential genes involved in a reaction. Thereafter, the distributions of the five categories of reaction-gene pairs from the predicted results have been compared with the distribution observed in experimental data for all the organisms using the Chi-Square Test (1% level of significance).

For *Leishmania donovani* and *Leishmania major*, the best model was selected based on the SSMSS score for the prediction of the essential reaction-gene combinations. These predicted reaction gene combinations were then classified into the five categories, like the other twelve species. The list of unique genes that were extracted from this predicted essential reaction-gene pairs was analyzed for their associated Gene Ontology (GO) terms [93,94] from the Uniprot database [95]. The percentages of genes associated with each GO term were calculated for both the organisms. Additionally, using the DAVID pathways enrichment tool [96], the essential genes were further analyzed to identify the significantly enriched KEGG pathways [97] that were associated with these essential genes.

Source codes of the entire machine learning strategy and pipeline are given in S1 Text, which consists, Training data set preparation and integration of heterogeneous features, Feature selection based on the space-filling concept, Dimension reduction using forced directed graph layout, and Semi-supervised classifier: LapSVM.

## 3. Results

### 3.1. Model validation with experimental data

The integrative proposed strategy (Fig 1) was applied and validated on twelve organisms (Table 1) with well-annotated genes essentiality information from experimental data obtained from the OGEE database [48].

### 3.2. Features frequently selected by the feature selection algorithm

The important features chosen by the feature selection algorithm have been represented in the heat map (See methods section for a detailed description of features and S1 Fig), where X-axis represents the name of the 82 features that have been selected at least once by the features selection algorithm and Y-axis corresponds to names of the organism. Red cell color indicates features selected by the feature selection algorithm in the corresponding organism. White-colored cell shows the feature that is not selected or is redundant. Among 289 features, three features, *viz.*, Reaction Network betweenness centrality (RN_betweenness), Reaction Network Page Rank centrality (RN_page_rank), and Flux Coupled Analysis Network Page Rank centrality (FCA_page_rank) are selected by the features selection algorithm for every organism. These frequently selected features are topological network features. Apart from these features, Information-theoretic features (Fourier sine or cosine coefficient, Mutual Information, Conditional Mutual Information) from nucleotide and peptide sequences are also selected. If a node is important in the reaction network and flux-coupled network, then there is a chance that the enzyme or protein which controls that particular reaction and its corresponding coding sequence is also essential.

### 3.3. Dimension reduction

After applying feature selection, the Kamada-Kawai dimension reduction technique [84] is used for visualization purposes. Here, a circular layout of each organism is observed. While the essential gene-reaction combinations are clustered together in one side of the arc in a 2-D circular layout, the non-essential reaction-gene combinations are clustered in the rest of the

circle. Now on applying Laplacian SVM, the classifier was able to easily classify gene essentiality based on their transformed 2-D feature and the limited label information. Now in different parameter combinations of Laplacian SVM, different trained models are obtained. To select the best model among trained models, the proposed SSMSS score has been used.

### 3.4. Robustness of the proposed score (SSMSS)

To check the robustness of the SSMSS score, the proposed strategy has been applied on both types of training data set (i.e., data set with 80–20% combination of samples and with the whole data set) for these twelve organisms. Using this 80% data points of the whole dataset, different types of training data set is further created with limited labeled data points in the range, i.e., i % Labeled (L) and (100—i%) Unlabeled (UL) data, where i = 1, 2, 3, 4, 5, 10, 30, 50, 70 and 90. In each category, labeled samples were chosen randomly from the master table. It is to be mentioned here that this selection of labeled data was conditionally randomized to ensure that both the essential and non-essential genes categories appear with equal probability. In this way, 100 data sets in each labeled category have been created. For the testing purpose, both the whole training data set and the 20% blind data set have been used for prediction. The parameters ($\sigma, \lambda, \gamma$) were tuned with four different values i.e. 0.01,0.1,1,10. Therefore, by tuning these model parameters using grid search generated 64 models for each data set have been created. After that, the prediction results were compared with the known gene essentiality information, which is publicly available from the experiment. Six supervised performance metrics have been calculated for the predicted class label with the known class label. After that, the association between the proposed score and auROC was assessed. To verify the linear relationship between auROC and the proposed score (SSMSS), the Pearson correlation coefficient has been calculated, and scatter plots were generated in different limited labeled data sets in each target organism (S2 Fig).

From the scatter plot (S2 Fig), it has been observed that in all the cases, Pearson correlation >0.75. Hence, it may be inferred that due to the linear relationship existing between auROC and the proposed score (SSMSS), the applicability of this scoring technique is asserted and can be used for the calculation of the performance measurement matrix and best model selection for the semi-supervised based classifier.

### 3.5. Predictive performance of the best models in the different labeled category on training and blind test data set

In a real-life scenario, only limited gene essentiality information is available for the less explored organisms. However, model building from this limited label data and determining how the highest score will select the best model is difficult. Hence, to test the model performance on known organisms by creating limited labeled datasets (i.e., by varying the limited labeled data from 1% to 90% from the 80% training dataset), six supervised performance metrics have been calculated for each category under different parameter combinations of $\sigma, \lambda, \gamma$ (See Section 2.5: The score for best model selection) for a detailed description of these parameters). Here, within each labeled category, the average behavior of the predictive performance (six supervised performance metrics) and the Score (SSMSS) of the best 100 trained models are plotted in Fig 2. This has been shown for two different conditions, training data set (80% of the whole data) and blind testing data set (20% of the whole data). As observed from the low standard deviations for each metrics (under each category), it is worth to mention that the accuracy for the training and testing are very similar in most of the cases.

From these plots, it has been observed that the model selection based on the SSMSS score in each category corresponds to a high auROC value of greater than 0.8 in all cases across all

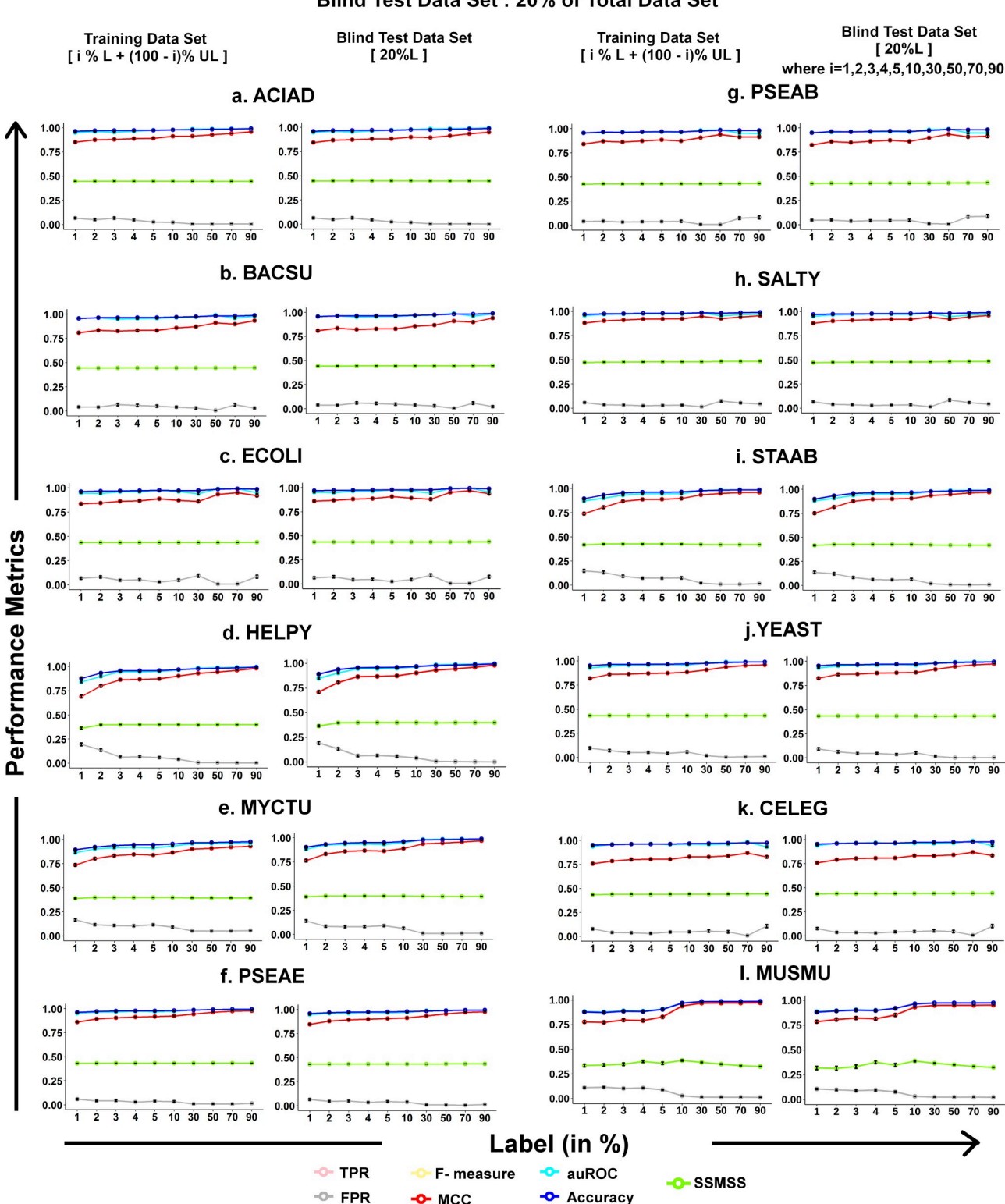

**Fig 2. Comparison of the predictive performance of the best models in the different labeled category.** The average performance of the best 100 models at training and blind testing for six supervised metrics (i.e., TPR, FPR, F-measure, MCC, auROC, accuracy) and SSMSS for each labeled type. The X-axis represents the category of labeled data, the Y-axis represents the value of performance metrics.

organisms. Also, it is observed that if the label increases, then model performance will also show higher accuracy. However, it is seen that the auROC score remains consistently high, using 1% labeled data or more, which establishes the fact that the proposed method can predict using a minimum of 1% labeled data. It has also been observed that this method is giving a consistent better predictive performance on both Prokaryotic and Eukaryotic organisms for both the data sets (80% training and 20% blind testing) and follow similar patterns for six supervised performance metrics in differently labeled categories. As the predictive performance of 20%, the blind data set is similar to training performance, so further, it can be concluded that model overfitting and underfitting is not arising in this case.

To compare the predictive performance of the proposed method, 1% labeled data set has been considered for each of the twelve organisms. For training, different supervised classifiers have been used, such as Random Forest [98], Naive Bayes [99], Logistic regression [100], J48 (C.45) Decision Tree [101] as well as our own previously reported Supervised essential gene prediction pipeline [40] on the whole dataset for testing (S3 Fig). In all of the cases, it is found that the proposed method performed better than all other methods using only 1% labeled data of the whole training dataset.

## 3.6. Effect of feature selection and dimension reduction in model performance

To compare the effect of feature selection and dimension reduction steps along with the LapSVM classifier, seven different types of classification scenarios, based on different dimension reduction technique such as PCA, MDS, FR, ICA, and KK, were simulated on training data set (80% data points) and blind testing (20% data points) data sets of twelve organisms. The corresponding performance was calculated on the blind test data set (Fig 3). Each training data set has only 1% labeled data, and the rest of them Unlabeled.

The seven scenarios were created with LapSVM classifier and combinations of features selection and dimension reduction techniques:

Scenario 1 (S1): Without feature selection and Without dimension reduction technique [WOFS +WODR]

Scenario 2 (S2): Without feature selection and With dimension reduction technique (Principal Component Analysis) [WOFS + DR (PCA)]

Scenario 3 (S3): Without feature selection and With dimension reduction technique (Metric Dimensional Scaling) [WOFS + DR (MDS)]

Scenario 4 (S4): Without feature selection and With dimension reduction technique (Fruchterman Reingold) [WOFS + DR (FR)]

Scenario 5 (S5): Without feature selection and With dimension reduction technique (Independent Component Analysis) [WOFS + DR (ICA)]

Scenario 6 (S6): Without feature selection and With dimension reduction technique (Kamada Kawai) [WOFS + DR (KK)]

Scenario 7 (S7): With feature selection (Unsupervised Feature Selection) and With dimension reduction technique (Kamada Kawai) [WFS (UFS) + DR (KK)]

From this analysis, it has been observed that for scenarios 1 to 5, the auROC value is very low, which signifies that dimension reduction techniques, e.g., PCA, MDS, FR, ICA, cannot significantly improve the gene essentiality prediction (Fig 3). On the other hand, for scenarios

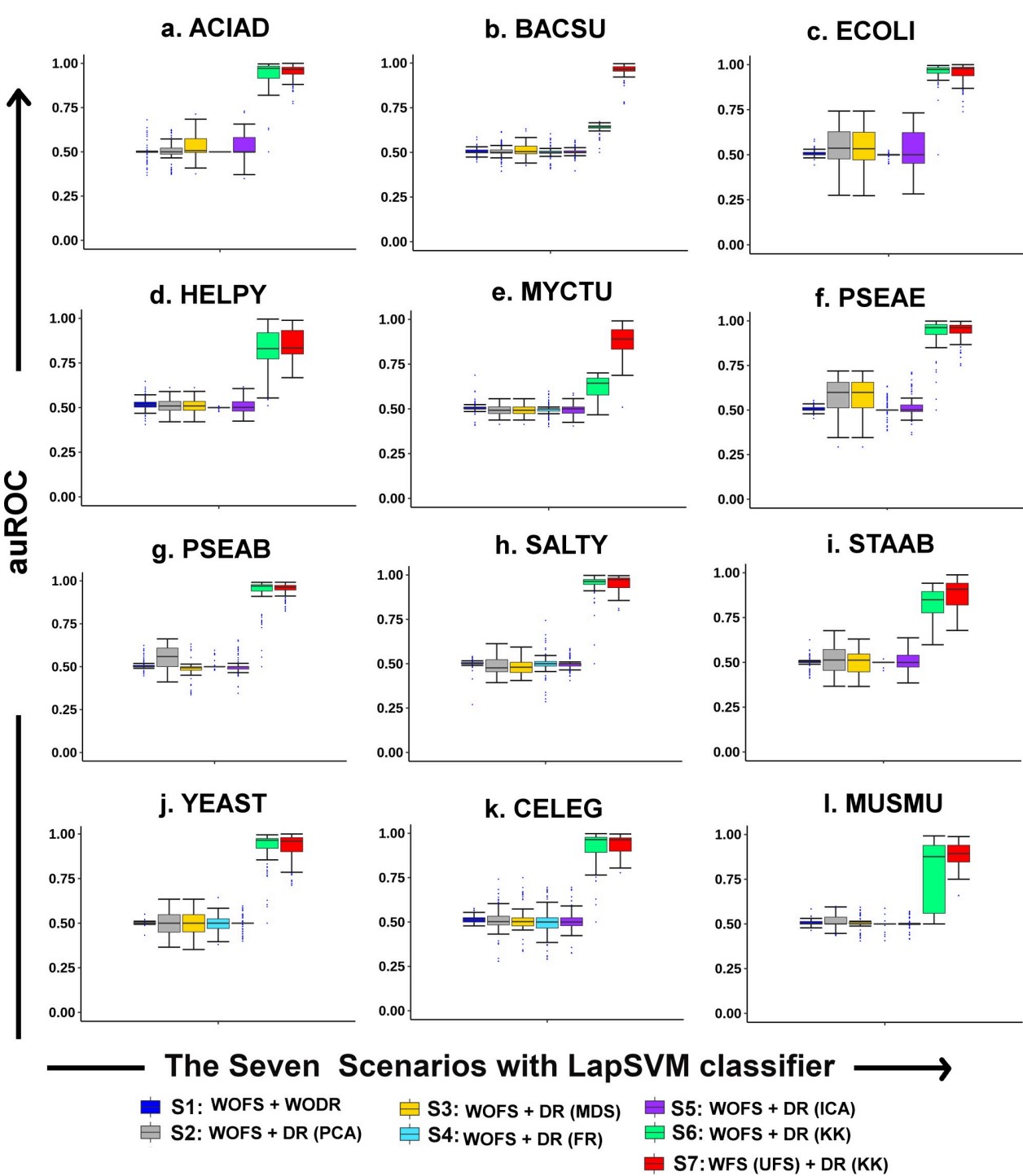

**Fig 3. Effect of feature selection and dimension reduction on model performance.** Comparison of the effect of different dimension reduction techniques PCA, MDS, FR, ICA, and KK (S2—S6) with S1 (Without Feature Selection and Without Dimension Reduction) and S7 (With Feature Selection and With Dimension Reduction-KK) when combined with LapSVM classifier. Plot represents the auROC value of 100 best models with 1% labeled data across all organisms.

6 and 7, it is observed that on applying the Kamada-Kawai method of dimension reduction along with unsupervised feature selection, the model performance (auROC) improves drastically in each target organism. On comparing the efficacy of Kamada-Kawai (KK) with the other dimension reduction methods using the one-tailed Mann-Whitney U Test, a significant improvement in auROC values ($P<0.01$) for all the twelve organisms was observed (S2 Table). Scenario 6 highlights the importance of this dimension reduction step, where it is found that even without feature selection, the dimension reduction step [S6: WOFS + DR (KK)] has a huge impact on the results ($P<0.01$) [S3 Table]. However, the feature selection step helped us in identifying the minimal set of features that contribute towards gene essentiality prediction with greater accuracy in all organisms (lower $P$-values obtained in Scenario 7 with [S7: WFS + DR (KK)]) (S3 Table). Hence, it is observed that the Kamada-Kawai dimension reduction technique, when combined with LapSVM, gives significantly better performance for all twelve organisms even when only 1% labeled data is used (Fig 3).

### 3.7. Predictive performance using whole training data set

In model organisms where gene essentiality information is sufficiently available at the genome-scale, blind testing can be applied. However, in less explored organisms where gene essentiality information is very less, a blind test cannot be applied as the reference size is very small. For these cases, the whole data set with limited labeled data can be used for model training and prediction purposes.

To establish the predictive performance of the proposed strategy on the whole training data set, 1% labeled data were selected randomly, and the remaining 99% data points were considered unlabeled for the twelve organisms, where the information of gene essentiality in genome-scale was available from the experiments. Now, this whole data set was trained by the proposed strategy. The best model was selected based on the highest score (SSMSS). The same data set is used for prediction from the best-trained model. The outcome of the proposed strategy can be visualized as three circles (Fig 4). The first circle represents the circular projection of the whole data set in 2-D after applying the Kamada Kawai dimension reduction technique with gene essentiality information from the experiment. The second circle shows the training data set with 1% labeled & 99% Unlabeled data and learning curve of the Laplacian model. The third circle shows the predicted gene essentiality label from the best-trained model. From Fig 4, it is observed that the proposed model also performed well (as similar circular patterns from experiment and predicted) on the whole training data set.

The predictive performance on both the data sets (80% and the Whole data set) has been compared by six supervised performance metrics (i.e., TPR, FPR, F-measure, MCC, auROC, and accuracy) based on actual and predicted labels from the proposed strategy. Here it has been observed that the average predictive performance of the 100 trained model with 80% data set is similar to the performance on the whole data set (S4 Fig).

### 3.8. Categorization of reaction-gene pairs

Categorization of the predicted essentiality information of reaction gene pairs into the five categories, viz. CEN, ME, MN, SE, and SN show that the distribution of reaction of the predicted results matches exactly with the distribution observed with the experimental data for each of the twelve organisms (Fig 5). Also, the Chi-square test was performed with a Null Hypothesis ($H_0$) that the two distributions of reaction (experimental vs. predicted) are similar for all twelve organisms. Here, it has been observed that the $P$-values of the Chi-square test ($P$-values are indicated in S4 Table) are greater than 0.01 in all the 12 organisms. As $P$-values are large, it can be concluded that the experimental distributions of reaction are not significantly different

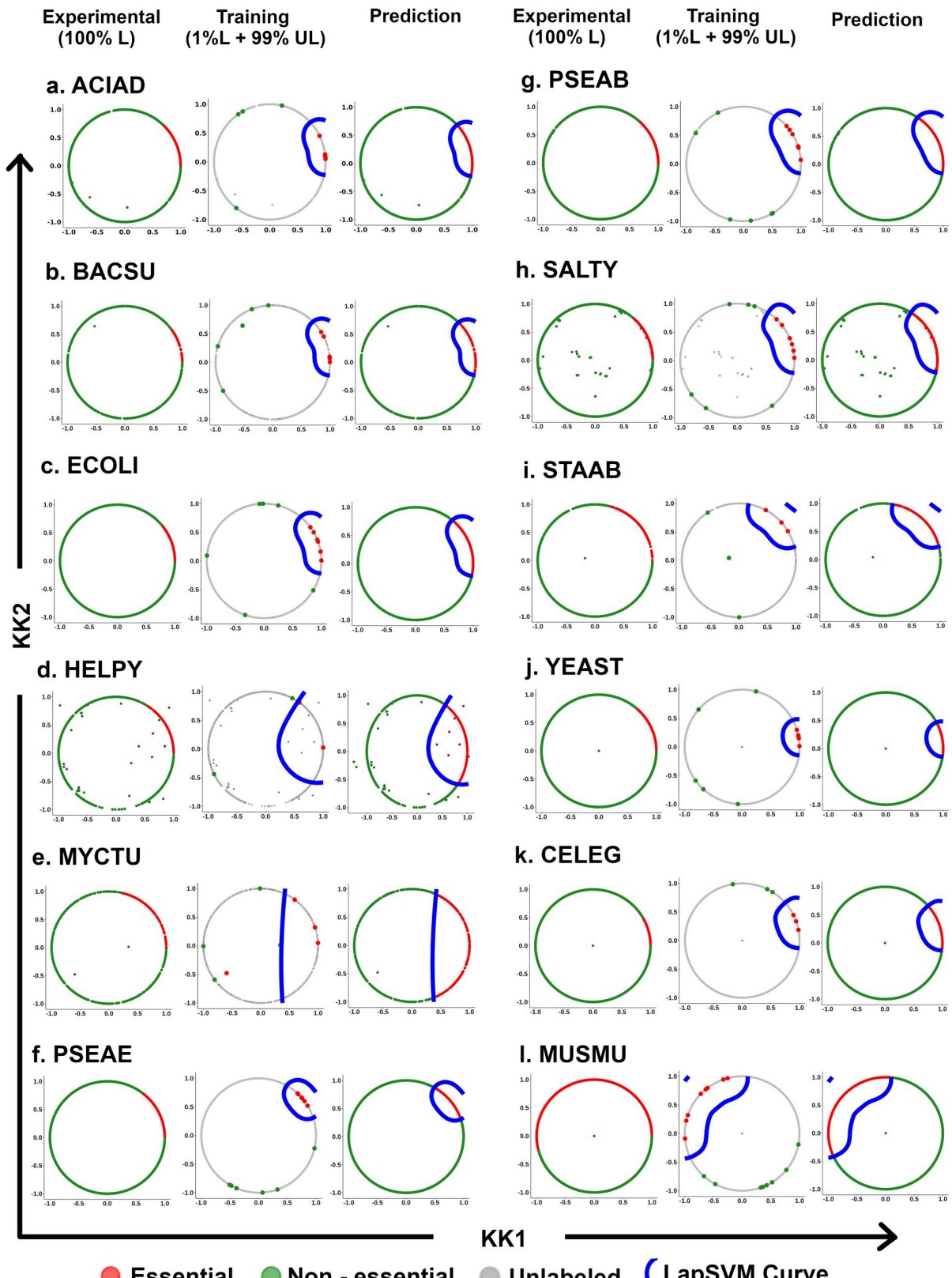

**Fig 4. Visualization of the outcome of the proposed strategy.** Essential, non-essential, and Unlabeled reaction gene pairs are colored accordingly Red, Green, and Gray. The learning curve for the best-trained model by LapSVM is colored with blue. The left circle represents the original data set with labeled data points. The middle circle shows the training data set with the learning curve, and the Right circle represents the prediction labeled with the learning curve.

from the predicted distributions. This pattern has been fairly consistent over all the organisms, where it is found that the highest fraction of reactions is regulated by single non-essential (red) or multiple non-essential genes (blue). On the other hand, fractions of reaction governed by a single essential gene are low due to a small number of minimally essential genes in all organisms. From this plot (Fig 5), it is also observed that the fractions of reactions governed by multiple essential genes are extremely low in each of the twelve organisms. These comprise the small set of reactions that are absolutely crucial for the survival of the organisms.

### 3.9. Case Study: *Leishmania donovani* and *Leishmania major*

The proposed strategy has been implemented for less explored organisms like *Leishmania donovani* (11 genes have genes essentiality information [102]) and *Leishmania major* (10 genes have genes essentiality information [102]) using the semi-supervised machine learning strategy. Here it is observed that the network centrality features and information-theoretic features, such as the Fourier cosine coefficient derived from the Kidera factor, have been selected by the feature selection algorithm in both the cases of *L. donovani* and *L. major*. Additionally, certain unique features were also selected for each of the two organisms (S1 Fig). When the Kamada-Kawai dimension reduction technique was applied on *Leishmania* data sets, a similar circular pattern was observed, like the other twelve organisms that helped the classifier in predicting gene essentiality (Fig 6A).

For the essential gene prediction, in the case of *Leishmania donovani*, 80 reaction-gene pairs were predicted as essential among 1129 reaction-gene pairs. For *Leishmania major*, 335 reaction-gene pairs were predicted as essential among 1188 reaction-gene pairs. The categorization of these reaction-gene pairs displayed a pattern similar to the distributions of reaction observed in the twelve model organisms (Fig 6B). Predicted gene essentiality information from the proposed pipeline is listed in (S5 and S6 Tables). The list of essential genes extracted from these reaction gene pairs consists of 44 essential genes of *L.donovani* and 194 of *L. major*.

These essential genes were associated with 53 and 219 Gene Ontology (Molecular Function) terms for *L. donovani* and *L. major*, respectively (S7 and S8 Tables). The Gene Ontology term that occurred most frequently with these essential genes were related to ATP binding in both the organisms. The pathway enrichment of these essential genes shows 11 significantly enriched KEGG pathways for *L. donovani* and 20 *L. major*. Although 8 KEGG pathways were found to be common among the two species, certain unique pathways specific to each species were also enriched for each of the two organisms (S9 and S10 Tables). Further experimental validation on these predicted results would confirm the role of these genes in these less-studied organisms.

## 4. Discussion

Essential gene prediction helps to unveil the complexities and survival strategies of many disease-causing organisms. The prediction of gene essentiality is a challenging task in machine learning due to the unavailability of sufficient experimentally labeled data and a proper metric for selection of the best model. Considering this limited gene essentiality information, the proposed pipeline has been able to predict gene essentiality at genome-scale using as small as a set of 1% labeled genes having gene essentiality information using both 80%-20% (training-blind

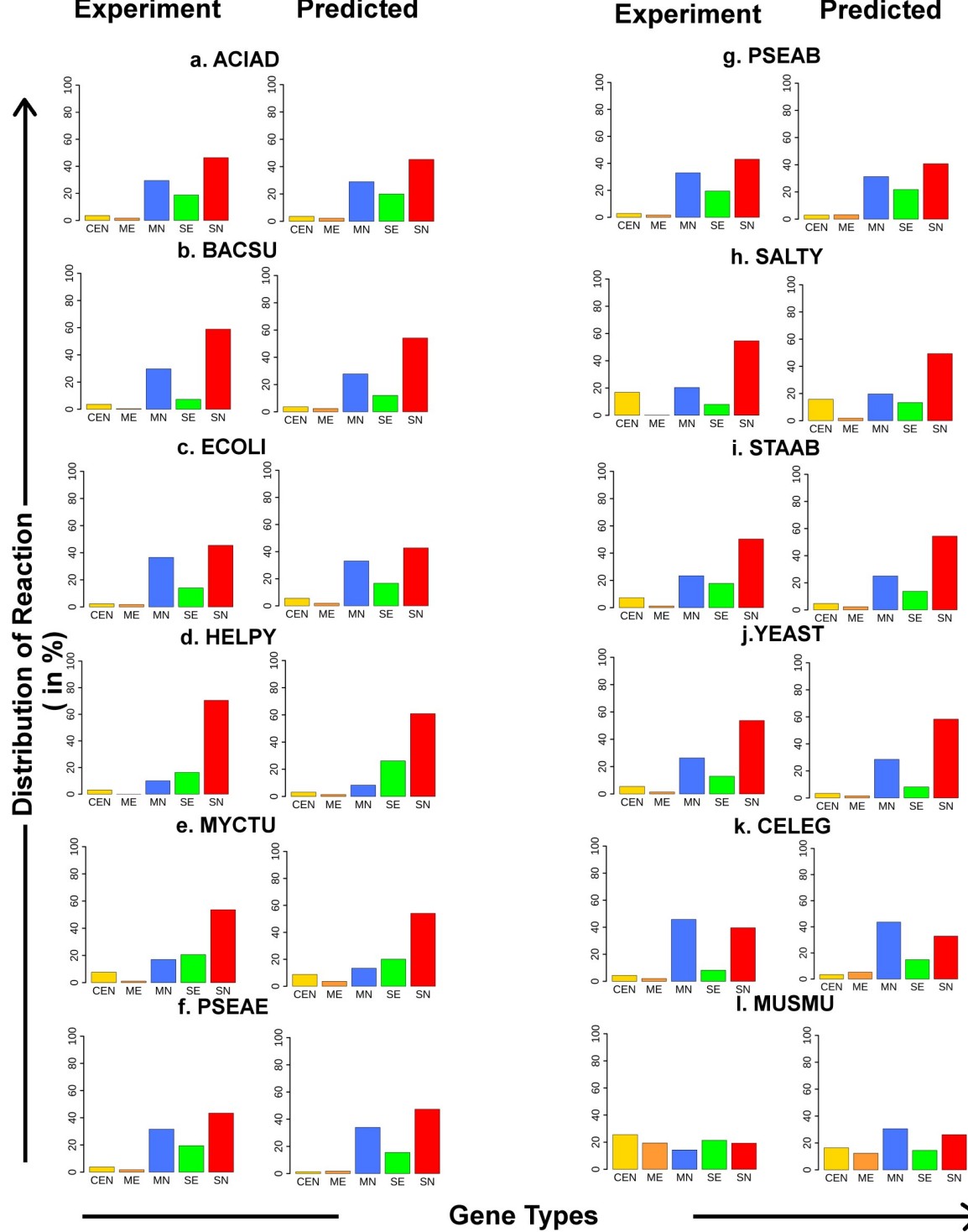

**Fig 5. Comparison of the distributions of reaction.** The reactions have been classified into five categories and the predicted distributions of reaction-gene pairs have been compared with the experimental data across all twelve organisms.

testing) dataset as well as the whole dataset for training and testing (Figs 2 and 4). This proposed pipeline consists of three key steps. First, the unsupervised feature selection algorithm

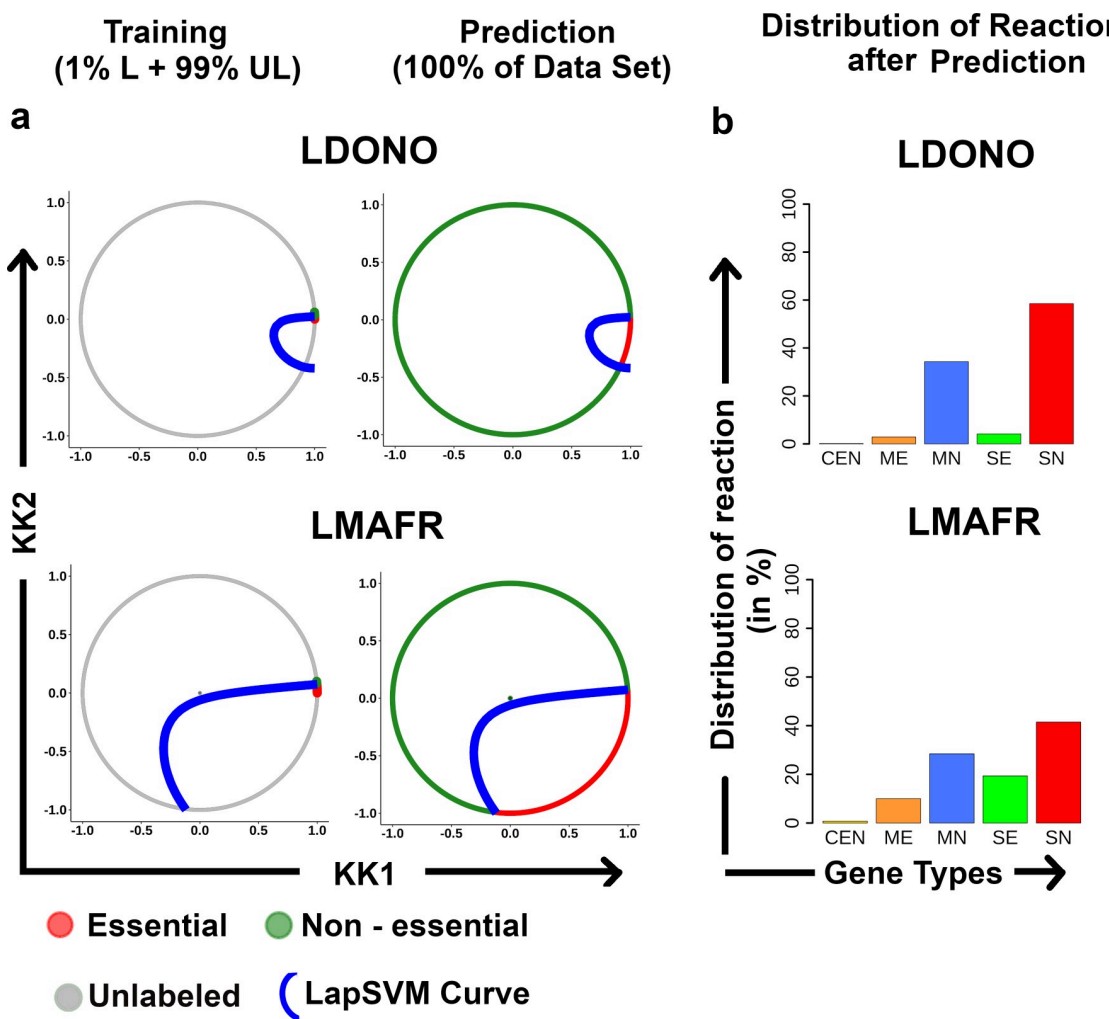

**Fig 6. Gene essentiality prediction in *L. donovani* and *L. major*. (a)** Kamada—Kawai dimension reduction on *Leishmania* datasets showed a circular pattern as observed for other organisms and the learning curve by LapSVM; **(b)** Distribution of reaction-gene pairs of *Leishmania* species into five categories.

has been used to select the relevant feature set from 289 feature set consisting of different heterogeneous biological features such as sequence-based features, and topological features derived from metabolic reaction network, and flux-coupled sub-network which help to distinguish between essential and non-essential reaction gene combinations. Here, it is observed that for every organism, the features selection algorithm selected three phenotypic features that have shown high correlation with gene essentiality, *viz.*, Reaction Network betweenness centrality (RN_betweenness), Reaction Network Page Rank centrality (RN_page_rank), and Flux Coupled Analysis Network Page Rank centrality (FCA_page_rank). Apart from these, novel features considered in this study, such as Information-theoretic features (Fourier sine coefficient and Fourier cosine coefficient derived from Kidera factor), were also correlated with gene essentiality prediction in most of the organisms. A distinguishing pattern between essential and non-essential genes for the selected features was captured by the feature selection algorithm, which helped the classifier to predict gene essentiality more accurately. Secondly, data set after feature selection was projected into a 2-D circular layout using the dimension reduction step Kamada-Kawai. This step is essential to project the high dimensional data into

a 2-D plane, which helps the classifier LapSVM to perform significantly better for all the organisms ($P<0.01$) (S2 Table). The results show that this dimension reduction step is capable of improving the prediction accuracy even without feature selection (Fig 3, S3 Table). However, we have also retained the feature selection step in our pipeline to identify the important features that are contributing to gene essentiality classification. After applying Kamada-Kawai, a distinct structured pattern was observed, showing the essential reaction-gene combinations clustered together and the non-essential reaction-gene combination in another cluster, each residing on the arc of a 2-D circular layout for each of the twelve known organisms (Fig 4). This clustered pattern of reaction-gene pairs helped the semi-supervised classifier (Laplacian SVM) build a non-linear curve that dissects this circle into essential and non-essential classes with significantly higher accuracy. The novelty of the proposed strategy lies in the integration of the Kamada-Kawai algorithm with the semi-supervised LapSVM classifier that contributes to the high accuracy obtained using the pipeline. This is evident from S2 Table, where a significantly higher model performance of the Kamada-Kawai step was observed over the other widely used dimension reduction techniques. Further, it has been observed that the LapSVM classifier, when combined with the Kamada-Kawai step, contributes to the higher predictive performance of this pipeline as compared to the other supervised machine learning techniques when only 1% labeled data is available (S3 Fig).

Thereafter, the SSMSS score was used to select the best model. Here it was observed that the selected model based on this scoring technique had a corresponding high auROC value when compared with the experimentally known labels (S2 Fig). This indicated the reliability of the proposed SSMSS score, which, although show high variation for less number of labeled data, is useful as an alternative score when the calculation of supervised metrics is difficult for best model selection.

After the successful validation of this strategy on twelve organisms, the methodology was used to annotate gene essentiality in less-studied organisms like *Leishmania donovani* and *Leishmania major*, for which less or no organism-specific machine learning studies are available. Here, it was observed that 80 reaction-gene pairs were predicted to be essential in *Leishmania donovani*. These reactions involved 44 genes that were mostly associated with ATP binding [GO:0005524], oxidoreductase activity [GO:0016491], and AMP deaminase activity [GO:0003876] GO terms. Similarly, in the case of *Leishmania major*, 335 reaction-gene pairs were predicted as essential that involve 194 genes. Here it is observed that in addition to the ATP binding and metal-ion binding activities [GO:0005524], some genes that were predicted to be essential were also associated with amino acid transmembrane transporter activity [GO:0015171], magnesium ion binding [GO:0000287], and protein serine/threonine kinase activity [GO:0004674] GO terms that were not observed in the *L. donovani*. On the other hand, in the case of *L. donovani*, the genes involved in flavin adenine dinucleotide binding [GO:0050660] and AMP deaminase activity [GO:0003876] were predicted as essential, which is not observed in *L. major*.

The KEGG pathway enrichment study performed on the essential gene sets of the two organisms–*L. donovani* and *L. major* throw light on the pathways that are crucial for the survival of these micro-organisms and can be considered as probable therapeutic targets. Here, it is observed that apart from the pathways involved in Purine metabolism, Pyrimidine metabolism, Pyruvate metabolism, etc., that were common to both the organisms, a set of unique pathways were also enriched in each of *L.major* and *L.donovani*. While in the case of *L. major*, the pathways involved in Glycolysis/Gluconeogenesis, Glycine, serine and threonine metabolism, Citrate cycle (TCA cycle), Pyruvate metabolism, and Inositol phosphate metabolism were significantly enriched ($P< 0.001$), the essential genes of *L. donovani* show a higher enrichment for Sphingolipid metabolism and Steroid biosynthesis pathways. Further, the

predicted essential reaction-gene combinations were categorized into five different groups (i.e., CEN, ME, MN, SE, and SN) that help to identify the individual reactions that are regulated by single or multiple essential genes. It may be mentioned here that a common pattern in these categories of distributions was observed across all the twelve organisms that corroborate well with the experimental observations (Fig 5). The Chi-Square Test performed to verify the difference in the experimental and predicted distributions showed no significant difference (S4 Table). A similar pattern was also predicted for *L. donovani* and *L. major* that further ascertains the validity of the predictions (Fig 6b). These results indicate the strength of the model in identifying true essential genes using a small amount labeled data, a selection of biologically relevant features to represent gene essentiality, and optimal parameters for curve formation to classify essential genes. The limitation of the proposed strategy is that, it requires the genome-scale reconstructed metabolic network, and at least 1% genes of this network should be annotated experimentally with gene essentiality information.

Using a graph-based semi-supervised machine learning scheme and combining different well-established methods in ML problems, a novel integrative approach has been proposed for essential gene prediction that shows universality in application to both prokaryotes and eukaryotes with limited labeled data. The run time of the pipeline is dependent on the size of the metabolic network (n), and the number of features (d) considered and can be represented as $T(n,d) = O(n^3 d^2)$. In the case of *L. major* and *L donovani*, the total runtime was 41 minutes and 48 minutes, respectively, when simulated on a workstation of Intel(R) Xeon(R) CPU E5-2620 v4 @ 2.10GHz with 32GB RAM. This strategy will provide experimental biologists a well standardized and validated methodology to predict gene essentiality of less-studied organisms as well as will cater to the theoretical scientists with a novel approach for binary classification problems when limited labeled data is available. The essential genes predicted using the pipeline provide important leads for the identification of novel therapeutic targets for antibiotic and vaccine development against disease-causing parasites, such as *Leishmania sp*.

## Supporting information

**S1 Fig. Heatmap plot of selected features by the feature selection algorithm.** Red cells indicate features selected by the feature selection algorithm in the corresponding organism. White cells show the feature that is not selected or is redundant.
(TIF)

**S2 Fig. Robustness evaluation of the proposed score (SSMSS).** Scatter plots is demonstrating an association between auROC and SSMSS in each labeled category data sets in different model parameters conditions for twelve organisms. The X-axis represents the score (SSMSS), and Y-axis represents the corresponding auROC. To represent each category, ten different colors are used.
(TIF)

**S3 Fig. Comparison of the predictive performance of the proposed strategy with other supervised methods.** Comparison of the performance of proposed strategy (PS) with supervised classifiers [i.e., Decision Tree (DT), Logistic regression (LR), Naive Bayes (NB), Random Forest (RF) and our own previously reported Supervised essential gene prediction pipeline] based on 1% labeled data on twelve organisms. The X-axis represents the different types of performance metrics for machine learning strategies, the Y-axis represents the value of performance metrics. Six different color codes were used to represent six different performance metrics.
(TIF)

**S4 Fig. Comparison of the predictive performance on both types of data sets (80% and whole data set).** Average predictive performance of the best 100 models on 80% training data set and performance of whole training data set containing the Limited Labeled (L = 1%) and remaining Unlabeled (UL) data for six supervised metrics (i.e., TPR, FPR, F-measure, MCC, auROC, accuracy) and SSMSS for each labeled type. The X-axis represents the different performance metrics, the Y-axis represents the value of performance metrics.
(TIF)

**S1 Table. List of curated 289 features.** List of curated 289 features for essential gene prediction.
(DOCX)

**S2 Table. Comparison of auROC of Kamada-Kawai (KK) dimension reduction technique with PCA, MDS, FR and ICA.** The values reported in the table represent the *P*-values obtained using the one-tailed Mann-Whitney U Test.
(DOCX)

**S3 Table. Comparison of the effect of feature selection and Kamada-Kawai (KK) dimension reduction technique on the model performance (auROC).** The values reported in the table represent the *P*-values obtained using the one-tailed Mann-Whitney U Test.
(DOCX)

**S4 Table. Comparison of percentage distribution of reaction into five categories from experiment vs predicted results.** The values reported in the table represent the *P*-values obtained using the Chi-square test.
(DOCX)

**S5 Table. Gene essentiality information of reaction gene combinations in *Leishmania donovani* predicted using the proposed pipeline.**
(DOCX)

**S6 Table Gene essentiality information of reaction gene combinations in Leishmania major predicted using the proposed pipeline.**
(DOCX)

**S7 Table. Gene Ontology (Molecular Function) terms of the predicted essential genes in *Leishmania donovani.***
(DOCX)

**S8 Table. Gene Ontology (Molecular Function) terms of the predicted essential genes in *Leishmania major.***
(DOCX)

**S9 Table. KEGG pathway enrichment of the predicted essential genes in *Leishmania donovani.***
(DOCX)

**S10 Table. KEGG pathway enrichment of the predicted essential genes in *Leishmania major.***
(DOCX)

**S1 Text. Source code of proposed machine learning strategy.** This supplementary text contains source code for the proposed machine learning strategy, including codes for (a) Training data set preparation and integration of heterogeneous features; (b) Feature selection based on

the space-filling concept; (c) Dimension reduction using forced directed graph layout; (d) Semi-supervised classifier LapSVM.
(DOCX)

## Acknowledgments

The authors acknowledge Dr. Leelavati Narlikar, Dr. Abhishek Subramanian, Mr. Kshitij Patil and Mr. Jarjish Rahaman for valuable suggestions and insightful comments.

## Author Contributions

**Conceptualization:** Ram Rup Sarkar.

**Data curation:** Sutanu Nandi.

**Formal analysis:** Sutanu Nandi, Piyali Ganguli.

**Investigation:** Sutanu Nandi.

**Methodology:** Sutanu Nandi.

**Supervision:** Ram Rup Sarkar.

**Validation:** Sutanu Nandi, Piyali Ganguli.

**Writing – original draft:** Sutanu Nandi.

**Writing – review & editing:** Piyali Ganguli, Ram Rup Sarkar.

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
