## [Decision Letter · Decision Letter 0]

24 Aug 2020

PONE-D-20-17655

Essential gene prediction using limited gene essentiality information– An Integrative Semi-supervised Machine Learning Strategy

PLOS ONE

Dear Dr. Sarkar,

Thank you for submitting your manuscript to PLOS ONE. After careful consideration, we feel that it has merit but does not fully meet PLOS ONE’s publication criteria as it currently stands. Therefore, we invite you to submit a revised version of the manuscript that addresses the points raised during the review process.

We look forward to receiving your revised manuscript.

Kind regards,

Seyedali Mirjalili

Academic Editor

PLOS ONE

Journal Requirements:

Sutanu Nandi acknowledges DST-INSPIRE for Senior Research Fellowship. Piyali Ganguli acknowledges CSIR for Senior Research Fellowship.

Reviewers' comments:

Reviewer's Responses to Questions

**Comments to the Author**

1. Is the manuscript technically sound, and do the data support the conclusions?

Reviewer #1: Yes

Reviewer #2: Yes

2. Has the statistical analysis been performed appropriately and rigorously? 

Reviewer #1: Yes

Reviewer #2: No

3. Have the authors made all data underlying the findings in their manuscript fully available?

Reviewer #1: Yes

Reviewer #2: Yes

4. Is the manuscript presented in an intelligible fashion and written in standard English?

Reviewer #1: Yes

Reviewer #2: Yes

5. Review Comments to the Author

Reviewer #1: This is a good work, but a number of major and minor amendments are required as follows:

* Potential applications of the proposed ML-based solution should be discussed

* There is no justification of the ML-based solution method. Why for this problem area, please discuss. There are many other similar methods in the literature in this area, so such a justification is required.

* There is no statistical test to judge about the significance of the ML-based solution’s results. Without such a statistical test, the conclusion cannot be supported.

* There is no discussion on the cost effectiveness of the proposed ML-based solution. What is the computational complexity? What is the runtime? Please include such discussions. You can also use the big oh notation to show the computation complexity.

* To have an unbiased view in the paper, there should be some discussions on the limitations of the proposed ML-based method

* Analysis of the results is missing in the paper. There is a big gap between the results and conclusion. There should be the result analysis between these two sections. After comparing the methods, you have to be able to analyse the results and relate them to the structure of all algorithms. It would be interesting to have your thoughts on why the method works that way? Such analyses would be the core of your work where you prove your understanding of the reason behind the results. You can also link the findings to the hypotheses of the paper. Long story short, this paper requires a very deep analysis from different perspectives

* How do you ensure that the comparison between ML-based and the comparative methods is fair?

* The proposed ML-based method might be sensitive to the values of its main controlling parameter. How did you tune the parameters?

* The main solution is an optimization, but the literature review of other metaheuristics is missing. Please provide an in-depth review to show readers a big picture of this field with recent and popular algorithms.

Some cosmetic comments:

* Avoid using first person.

* Highlights are missing.

* Avoid using abbreviations and acronyms in title, abstract, headings and highlights.

* Please avoid having heading after heading with nothing in between, either merge your headings or provide a small paragraph in between.

* Abstract is too short. Abstract should have one sentence per each: context and background, motivation, hypothesis, methods, results, conclusions.

* The first time you use an acronym in the text, please write the full name and the acronym in parenthesis. Do not use acronyms in the title, abstract, chapter headings and highlights.

* The results should be further elaborated to show how they could be used for the real applications.

* The originality of the paper needs to be further clarified.

Reviewer #2: The authors have developed a new ML pipeline to help in the annotation of essential genes of less explored disease-causing organisms for which very limited experimental data is available. The proposed strategy combines unsupervised feature selection technique, dimension reduction using the Kamada-Kawai algorithm, and semi-supervised ML algorithm employing Laplacian SVM, for prediction of essential and non-essential genes from genome-scale metabolic networks using very limited labeled dataset. A novel scoring technique SSMSS, equivalent to auROC, has been proposed for the selection of best model when supervised performance metrics calculation is difficult due to lack of data. However, there are few things need to be addressed as a way to improve the quality of the presented work, some of the suggested comments are listed as follows:

1. There is no justification of the method. Why for this problem area, please discuss. There are many other similar methods in the literature in this area, so such a justification is required.

2. There is no statistical test to judge about the significance of the method’s results. Without such a statistical test, the conclusion cannot be supported.

3. The main solution is an ML based, but the literature review of other ML methods is missing. Please provide an in-depth review to show readers a big picture of this field with recent and popular algorithms.

4. The given discussion on the obtained results should be further improved, for example, there were some noticeable performance improvements, but the reasons behind are not cohesively discussed, improvements were presented as a rule of thumb. The authors should make linkage between the achieved improvement and the proposed methodology to make kind of justifications on why the results are significant?

5. English proofread is required by authors before the final submission.

6. PLOS authors have the option to publish the peer review history of their article (what does this mean?). If published, this will include your full peer review and any attached files.

Reviewer #1: No

Reviewer #2: No

---

## [Author Response · Author response to Decision Letter 0]

23 Sep 2020

Response to Reviewers

To,

Dr. Seyedali Mirjalili

Academic Editor

PLOS ONE

Dear Dr. Mirjalili,

First, we would like to sincerely thank you for giving us the opportunity to perform the revision of our manuscript. We also thank both the Reviewers for a detailed and critical review of our manuscript, highlighting its deficiencies along with indicating the usefulness of our endeavour. The comments are insightful and have brought out several points that were not clearly stated in the earlier version of the paper. We are also thankful to the Reviewers for recognizing the importance of our work and effort and advising us to submit the revised version of the manuscript. The manuscript is now thoroughly revised following the suggestions/comments made by the Reviewers (Highlighted in the revised Manuscript, Marked-up Copy), and the point to point answers are given below.

We hope this revised version will satisfy you and the reviewers and will be acceptable to the journal.

REPLY TO ACADEMIC EDITOR’S COMMENTS

Comment 1: Please ensure that your manuscript meets PLOS ONE's style requirements, including those for file naming. 

Reply: We have now reformatted the manuscript (Font Style, Font Size, Line Spacing, Figure Captions, etc.) according to the guidelines and style requirements of PLOS ONE. We have also formatted the references using the PLOS ONE format with Mendeley Desktop (Version-1.19.4). However, we have retained the section numbers for the ease of cross referencing. 

Comment 2: Please remove any funding-related text from the manuscript and let us know how you would like to update your Funding Statement.

Reply: We have now removed the funding statement from the Acknowledgement Section. The revised Funding Statement is now provided here as well as in the Cover Letter.

Funding Statement: We thank SERB, Department of Science and Technology, Govt. of India (DST/ICPS/EDA/2018) and DBT, Department of Biotechnology, Govt. of India (File No. BT/PR14958/BID/7/537/2015), for providing financial support to Ram Rup Sarkar. Sutanu Nandi acknowledges DST-INSPIRE for Senior Research Fellowship. Piyali Ganguli acknowledges the Council of Scientific & Industrial Research (CSIR) for the Senior Research Fellowship. The funders had no role in study design, data collection and analysis, decision to publish, or preparation of the manuscript.

REPLY TO REVIEWERS’ COMMENTS

Reviewer #1: This is a good work, but a number of major and minor amendments are required as follows:

Reply: We thank the Reviewer for appreciating our efforts. We have incorporated the changes in the current version of the manuscript according to the suggestions. We hope that this new version of the manuscript will answer all the concerns raised by the Reviewer.

Comment 1: Potential applications of the proposed ML-based solution should be discussed

Reply: The potential application of the pipeline has been discussed both in the Abstract (Revised Manuscript, Line # 32-35), Introduction (Revised Manuscript, Line # 155-160), and Discussion (Revised Manuscript, Line # 763-768).

Comment 2: There is no justification of the ML-based solution method. Why for this problem area, please discuss. There are many other similar methods in the literature in this area, so such a justification is required.

Reply: We thank the Reviewer for the useful suggestion. We have now incorporated a detailed review of all the computational methods available in the literature that have been used to study essential genes (Revised Manuscript, Line # 47-90, highlighted in the manuscript). Here, we have also included a review of the ML-based solutions and justified its necessity for essential gene prediction.

Comment 3: There is no statistical test to judge about the significance of the ML-based solution’s results. Without such a statistical test, the conclusion cannot be supported.

Reply: According to the suggestion of the Reviewer, we have now included three supplementary tables (Table S2, S3, and S4) to show the statistical significance of our results when compared with other methods. In Table S2, we have shown the comparison of auROC of the Kamada-Kawai (KK) dimension reduction technique as used in our pipeline with other dimension reduction techniques such as PCA, MDS, FR, and ICA, using the one-tailed Mann-Whitney U Test. Similarly, in Table S3, we have shown the comparison of the effect of feature selection and Kamada-Kawai (KK) dimension Reduction technique on the model performance (auROC). Here again, the P-values have been computed using the one-tailed Mann-Whitney U Test. On the other hand, in Table S4, we have shown a comparison of the percentage distribution of the reactions into five categories as obtained from the experiments vs. our predicted results. The P-values reported in this table have been obtained using the Chi-square test.

The corresponding results have also been discussed in the current revised version of the manuscript (Revised Manuscript, Line # 346-348, 428-431, 566-574, 620-625, 696, 747-749, Highlighted)

Comment 4: There is no discussion on the cost effectiveness of the proposed ML-based solution. What is the computational complexity? What is the runtime? Please include such discussions. You can also use the big oh notation to show the computation complexity.

Reply: The computational time complexity of the proposed essential gene prediction pipeline has now been included in the Methods section (Revised Manuscript, Section 2.6) of the manuscript (Revised Manuscript, Line # 402-416). The runtime of the use cases L. major and L. donovani have also been mentioned in the Discussion section (Revised Manuscript, Line # 759-763).

Comment 5: To have an unbiased view in the paper, there should be some discussions on the limitations of the proposed ML-based method

Reply: The limitations of the pipeline are that it requires the genome-scale reconstructed metabolic network of the organism, and at least 1% genes of this network should be annotated experimentally with gene essentiality information. The limitations have now been discussed in the revised version of the manuscript (Revised Manuscript, Line # 753-755).

Comment 6: Analysis of the results is missing in the paper. There is a big gap between the results and the conclusion. There should be the result analysis between these two sections. After comparing the methods, you have to be able to analyze the results and relate them to the structure of all algorithms. It would be interesting to have your thoughts on why the method works that way? Such analyses would be the core of your work where you prove your understanding of the reason behind the results. You can also link the findings to the hypotheses of the paper. Long story short, this paper requires a very deep analysis from different perspectives

Reply: According to the suggestion, we have now revised our Discussion section and analyzed the results in detail. We have now explained the different components of our pipeline and how each is contributing towards enhancing the accuracy of our predictions (Revised Manuscript, Line # 694-711).

Comment 7: How do you ensure that the comparison between ML-based and the comparative methods is fair?

Reply: We apologize for the lack of clarity in our previous version which have given rise to the confusion. In Figure 3, we have tried to compare the effect of different dimension reduction techniques when combined with feature selection and LapSVM classifier. To ensure a fair comparison between the methods, we have used experimentally known dataset (gold standard) from all the twelve organisms and have also shown the statistical significance of the improvement in the auROC values when compared with the proposed method (Table S2, S3). Additionally, we have also revised the x-axis label of Figure 3 to ensure clarity of our objective.

In Supplementary Figure S3, we have also shown a comparison of the proposed method with other supervised ML techniques using the same dataset to ensure a fair comparison of all the methods using the default parameters.

Comment 8: The proposed ML-based method might be sensitive to the values of its main controlling parameter. How did you tune the parameters?

Reply: The proposed pipeline is sensitive to the three parameters (kernel parameter [Radial Basis Function (RBF) kernel parameter sigma ( )] and LapSVM parameters [lambda ( ): L2 regularization parameter and gamma ( ): the weight of the unlabeled data]) which have been tuned using the grid search technique with four different values, i.e., 0.01,0.1,1,10. As a result, 64 models were created for each dataset using a different combination of the parameter values. This has already been explained in the Methods section (section 2.5) Lines 388-395 of the manuscript. Hope this answers the question raised by the Reviewer. 

Comment 9: The main solution is an optimization, but the literature review of other metaheuristics is missing. Please provide an in-depth review to show readers a big picture of this field with recent and popular algorithms.

Reply: We appreciate the suggestion from the Reviewer. We have now included a detailed review of all the recent and popular meta-heuristic techniques that have been applied for the optimization of the hyper-parameter of ML algorithms. The corresponding review have also been discussed in the current revised version of the manuscript (Revised Manuscript, Line # 73-90, Highlighted)

Comment 10: Some cosmetic comments:

Reply: We thank the Reviewer for providing a critical review of the manuscript and indicating the errors in the first version. We have now corrected all the mistakes according to the suggestions and hope that the revised version will answer all the queries raised. 

* Avoid using first person. 

Reply: The current version has now been revised without the use of first person. 

* Highlights are missing.

Reply: According to the best of our knowledge, the journal has no provision for providing Highlights separately. However, we have tried to highlight the key findings and applications of our work in the concluding paragraph of the Discussion. 

* Avoid using abbreviations and acronyms in title, abstract, headings and highlights.

Reply: We have made sure to avoid the use of acronyms in title, abstract, and headings as far as possible in the current version.

* Please avoid having heading after heading with nothing in between, either merge your headings or provide a small paragraph in between.

Reply: We have now added a few lines in between subsequent headings.

* Abstract is too short. Abstract should have one sentence per each: context and background, motivation, hypothesis, methods, results, conclusions.

Reply: To the best of our knowledge, the journal format requires an unstructured abstract within 300 words limit. Hence, we are unable to elaborate it any further. However, according to the suggestion of the Reviewer, we have tried to include background, objective, methods, results, and conclusions briefly within the specified word limit.

* The first time you use an acronym in the text, please write the full name and the acronym in parenthesis. Do not use acronyms in the title, abstract, chapter headings and highlights.

Reply: We have made sure to avoid the use of acronyms in the title, abstract, chapter headings, and highlights as far as possible in the current version.

* The results should be further elaborated to show how they could be used for the real applications.

Reply: The necessity of the essential gene prediction has been discussed in the introduction (Revised Manuscript, Line #39-43). We have also discussed the usefulness and application of our pipeline in the study of essential genes of organisms with limited gene essentiality information in the manuscript (Revised Manuscript, Line # 155-160, 763-768). 

* The originality of the paper needs to be further clarified.

Reply: In this work, we have tried to predict the essential genes of an organism with only 1% labeled data with high accuracy. To the best of our knowledge and the current literature review, this work is the first study where gene essentiality has been predicted with such limited experimental data and the first pipeline to show good efficacy for both prokaryotes and eukaryotes. This has been explained in the Discussion (Revised Manuscript, Line # 673-678)

Reviewer #2: The authors have developed a new ML pipeline to help in the annotation of essential genes of less explored disease-causing organisms for which very limited experimental data is available. The proposed strategy combines unsupervised feature selection technique, dimension reduction using the Kamada-Kawai algorithm, and semi-supervised ML algorithm employing Laplacian SVM, for prediction of essential and non-essential genes from genome-scale metabolic networks using very limited labeled dataset. A novel scoring technique SSMSS, equivalent to auROC, has been proposed for the selection of best model when supervised performance metrics calculation is difficult due to lack of data. However, there are few things need to be addressed as a way to improve the quality of the presented work, some of the suggested comments are listed as follows:

Reply: We thank the Reviewer for the critical review and the useful suggestions. In the revised version of the manuscript, we have now tried to answer the queries raised by the Reviewer to enhance the clarity of our manuscript.

Comment 1: There is no justification of the method. Why for this problem area, please discuss. There are many other similar methods in the literature in this area, so such a justification is required.

Reply: A detailed review of all the computational methods used to study essential genes have now been included in the revised version of the manuscript (Revised Manuscript, Line #48-90, highlighted in the manuscript). Alongside this, we have also included a detailed review of the existing ML-based solutions and justified the necessity of our pipeline for the essential gene prediction. 

Comment 2: There is no statistical test to judge about the significance of the method’s results. Without such a statistical test, the conclusion cannot be supported.

Reply: Three supplementary tables (Table S2, S3, and S4) have now been included to show the statistical significance of our results and support our conclusions using Mann Whitney U Test and Chi-Square Test. In Table S2, we have shown the comparison of auROC of the Kamada-Kawai (KK) dimension Reduction technique as used in our pipeline with dimension reduction techniques such as PCA, MDS, FR, and ICA, using the one-tailed Mann-Whitney U Test. Similarly, in Table S3, we have shown the comparison of the effect of feature selection and Kamada-Kawai (KK) dimension Reduction technique on the model performance (auROC). Here again, the P-values have been computed using the one-tailed Mann-Whitney U Test. On the other hand, in Table S4, we have shown a comparison of the percentage distribution of the reactions into five categories as obtained from the experiments vs. our predicted results. The P-values reported in this table have been obtained using the Chi-square test.

The corresponding results have also been discussed in the current revised version of the manuscript (Revised Manuscript, Lines # 346-348, 428-431, 566-574, 620-625, 696, 747-749, Highlighted)

Comment 3: The main solution is an ML-based, but the literature review of other ML methods is missing. Please provide an in-depth review to show readers a big picture of this field with recent and popular algorithms.

Reply: We have now incorporated the detailed recent and popular ML algorithms for essential gene prediction to show the big picture of the field in the Introduction (Revised Manuscript, Line # 48-90). We hope this answers the concern raised by the Reviewer.

Comment 4: The given discussion on the obtained results should be further improved, for example, there were some noticeable performance improvements, but the reasons behind are not cohesively discussed, improvements were presented as a rule of thumb. The authors should make linkage between the achieved improvement and the proposed methodology to make kind of justifications on why the results are significant?

Reply: According to the suggestion of the Reviewer, we have now analyzed the results in detail in the discussion section. Here, we have now explained the different components of our pipeline and discussed how each is contributing towards enhancing the performance and accuracy of our predictions (Revised Manuscript, Line # 694-711). The statistical test performed to analyze the improvements in our results due to the incorporation of the dimension reduction and feature selection steps have also been shown to justify our conclusions.

Comment 5: English proofread is required by authors before the final submission.

Reply: The typos, grammar, and punctuation of the manuscript have now been thoroughly revised and corrected. Kindly note that, apart from manual checking, we have also used the Grammarly software (https://www.grammarly.com/edu; License and access provided to our Institute Library, CSIR-National Chemical Laboratory through NKRC consortia) and the changes are incorporated in the revised version of the manuscript to enhance the readability of the paper.

We hope that this revised version will satisfy the reviewers and will be acceptable for publication in PLoS ONE

---

## [Decision Letter · Decision Letter 1]

30 Oct 2020

PONE-D-20-17655R1

Essential gene prediction using limited gene essentiality information– An Integrative Semi-supervised Machine Learning Strategy

PLOS ONE

Dear Dr. Sarkar,

Thank you for submitting your manuscript to PLOS ONE. After careful consideration, we feel that it has merit but does not fully meet PLOS ONE’s publication criteria as it currently stands. Therefore, we invite you to submit a revised version of the manuscript that addresses the points raised during the review process.

We look forward to receiving your revised manuscript.

Kind regards,

Seyedali Mirjalili

Academic Editor

PLOS ONE

Reviewers' comments:

Reviewer's Responses to Questions

**Comments to the Author**

1. If the authors have adequately addressed your comments raised in a previous round of review and you feel that this manuscript is now acceptable for publication, you may indicate that here to bypass the “Comments to the Author” section, enter your conflict of interest statement in the “Confidential to Editor” section, and submit your "Accept" recommendation.

Reviewer #1: (No Response)

Reviewer #3: (No Response)

2. Is the manuscript technically sound, and do the data support the conclusions?

Reviewer #1: (No Response)

Reviewer #3: (No Response)

3. Has the statistical analysis been performed appropriately and rigorously? 

Reviewer #1: (No Response)

Reviewer #3: Yes

4. Have the authors made all data underlying the findings in their manuscript fully available?

Reviewer #1: (No Response)

Reviewer #3: Yes

5. Is the manuscript presented in an intelligible fashion and written in standard English?

Reviewer #1: (No Response)

Reviewer #3: Yes

6. Review Comments to the Author

Reviewer #1: My comments have been addressed. My comments have been addressed. My comments have been addressed. My comments have been addressed. My comments have been addressed.

Reviewer #3: In this paper, the authors proposed the machine learning pipeline for essential gene prediction. This is a good work. The quality of the paper has been improved after revision. There are several observations listed as follows:

1. Please separate the Introduction into two sections (1) Introduction and (2) Related Works.

2. Authors performed the feature selection and then used the dimensionality reduction method. Why use both feature selection and dimensionality reduction techniques in the same model? You can reduce a great number of features with only feature selection or dimensionality reduction. The justification for using both technique is required.

3. In Figure 2, the results for the training and testing accuracy are similar in most cases. Have you run the experiment for several times (with different data partition) and then take the average?

7. PLOS authors have the option to publish the peer review history of their article (what does this mean?). If published, this will include your full peer review and any attached files.

Reviewer #1: No

Reviewer #3: No

---

## [Author Response · Author response to Decision Letter 1]

4 Nov 2020

Response to Reviewers

To,

Dr. Seyedali Mirjalili

Academic Editor

PLOS ONE

Dear Dr. Mirjalili,

First, we would like to sincerely thank you for giving us another opportunity to perform the revision of our manuscript. We also thank the Reviewers for their useful suggestions and recognizing the importance of our work and effort. The manuscript has now been revised further following the suggestions/comments made by the third Reviewer (Highlighted in the revised Manuscript, Marked-up Copy), and the point to point answers are given below.

We hope this revised version will satisfy you and the reviewers and will be acceptable to the journal.

REPLY TO REVIEWERS’ COMMENTS

Reviewer #1: My comments have been addressed. 

Reply: We thank the Reviewer for appreciating our efforts. 

Reviewer #3: In this paper, the authors proposed the machine learning pipeline for essential gene prediction. This is a good work. The quality of the paper has been improved after revision. There are several observations listed as follows:

Reply: We thank the Reviewer for the critical review and the useful suggestions. In the revised version of the manuscript, we have now tried to answer the queries raised by the Reviewer to enhance the clarity of our manuscript. 

Comment 1: Please separate the Introduction into two sections (1) Introduction and (2) Related Works.

Reply: We thank the reviewer for the useful suggestion. We admit that the current length of the introduction is a bit on the higher side. However, the manuscript organization according to the Plos One format does not accommodate any separate section for Related Works. This might affect the flow of the introduction for the readers and might break its continuity with the subsequent sections. Hence, we have retained the introduction in its current form. However, if the editorial board finds it appropriate and give permission, we are willing to change the format for the introduction section, if required. 

Comment 2: Authors performed the feature selection and then used the dimensionality reduction method. Why use both feature selection and dimensionality reduction techniques in the same model? You can reduce a great number of features with only feature selection or dimensionality reduction. The justification for using both technique is required.

Reply: Both the feature selection and the dimensionality reduction methods are used for reducing the number of features in a dataset. Feature selection is also used for selecting the relevant features without changing the original values, whereas, the dimensionality reduction step transforms the higher dimensional features into a lower dimension. It is very difficult to identify the key features from the dimension reduction technique which are contributing for classification.

From our analysis (Fig. 3), we observe that in our pipeline, the Dimension Reduction technique Kamada Kawai is contributing heavily towards enhancing the model performance for all the organisms. However, we have also retained the Feature Selection step to identify the important features that are contributing to gene essentiality classification. 

The corresponding results have also been discussed in the current revised version of the manuscript (Revised Manuscript, Lines #348-354,719-721, Highlighted)

Comment 3: In Figure 2, the results for the training and testing accuracy are similar in most cases. Have you run the experiment for several times (with different data partition) and then take the average?

Reply: We apologize for the lack of clarity of the Figure 2 which has arisen due to the scaling of the Y-axis. In Figure 2, although the accuracy for the training and testing looks similar in most cases, the values are not exactly the same. 

To establish the model consistency and reproducibility, the model has been run by creating different data sets in each labeled/partition category (e.g. 1% category, where 100 datasets of 1% labelled category were created randomly from the master table). It is to be mentioned here that this selection of labeled data was conditionally randomized to ensure that both the essential and non-essential genes categories appear with equal probability. Thereafter the model performance of these 100 models were represented by plotting the average and standard deviation of each metrics. As observed from the low standard deviations for each metrics (under each category), it is worth to mention that the accuracy for the training and testing are very similar in most of the cases.

It is to be noted that 100 datasets were created for each of the 10 partition categories (1%, 2%, 3%, 4%, 5%, 10%, 30%, 50%, 70% and 90%). Hence the total number of datasets for each organism for training was 100x10=1000 datasets. (Revised Manuscript, Line # 523-529). We hope this answers the concern raised by the Reviewer.

We hope that this revised version will satisfy the reviewers and will be acceptable for publication in PLoS ONE

---

## [Decision Letter · Decision Letter 2]

12 Nov 2020

Essential gene prediction using limited gene essentiality information– An Integrative Semi-supervised Machine Learning Strategy

PONE-D-20-17655R2

Dear Dr. Sarkar,

We’re pleased to inform you that your manuscript has been judged scientifically suitable for publication and will be formally accepted for publication once it meets all outstanding technical requirements.

Kind regards,

Seyedali Mirjalili

Academic Editor

PLOS ONE

Additional Editor Comments (optional):

Reviewers' comments:

Reviewer's Responses to Questions

**Comments to the Author**

1. If the authors have adequately addressed your comments raised in a previous round of review and you feel that this manuscript is now acceptable for publication, you may indicate that here to bypass the “Comments to the Author” section, enter your conflict of interest statement in the “Confidential to Editor” section, and submit your "Accept" recommendation.

Reviewer #1: (No Response)

Reviewer #3: All comments have been addressed

2. Is the manuscript technically sound, and do the data support the conclusions?

Reviewer #1: (No Response)

Reviewer #3: (No Response)

3. Has the statistical analysis been performed appropriately and rigorously? 

Reviewer #1: (No Response)

Reviewer #3: (No Response)

4. Have the authors made all data underlying the findings in their manuscript fully available?

Reviewer #1: (No Response)

Reviewer #3: (No Response)

5. Is the manuscript presented in an intelligible fashion and written in standard English?

Reviewer #1: (No Response)

Reviewer #3: Yes

6. Review Comments to the Author

Reviewer #1: My comments have been addressed, so I recommend acceptance. .

Reviewer #3: In the revised paper, the authors have addressed all my concerns.

7. PLOS authors have the option to publish the peer review history of their article (what does this mean?). If published, this will include your full peer review and any attached files.

Reviewer #1: No

Reviewer #3: No

---

## [Editor Report · Acceptance letter]

17 Nov 2020

PONE-D-20-17655R2 

Essential gene prediction using limited gene essentiality information– An Integrative Semi-supervised Machine Learning Strategy 

Dear Dr. Sarkar:

I'm pleased to inform you that your manuscript has been deemed suitable for publication in PLOS ONE. Congratulations! Your manuscript is now with our production department. 

Kind regards, 

on behalf of

Prof. Seyedali Mirjalili 

Academic Editor

PLOS ONE